# dKV-Cache: The Cache for Diffusion Language Models

**Xinyin Ma   Runpeng Yu   Gongfan Fang   Xinchao Wang**[*]
National University of Singapore
maxinyin@u.nus.edu, xinchao@nus.edu.sg

## Abstract

Diffusion Language Models (DLMs) have been seen as a promising competitor for autoregressive language models (ARs). However, diffusion language models have long been constrained by slow inference. A core challenge is that their non-autoregressive architecture and bidirectional attention preclude the key–value cache that accelerates decoding. We address this bottleneck by proposing a KV-cache-like mechanism, **d**elayed **KV-Cache**, for the denoising process of DLMs. Our approach is motivated by the observation that different tokens have distinct representation dynamics throughout the diffusion process. Accordingly, we propose a delayed and conditioned caching strategy for key and value states. We design two complementary variants to cache key and value step-by-step: (1) dKV-Cache-Decode, which provides almost lossless acceleration, and even improves performance on long sequences, suggesting that existing DLMs may under-utilise contextual information during inference. (2) dKV-Cache-Greedy, which has aggressive caching with reduced lifespan, achieving higher speed-ups with quadratic time complexity at the cost of some performance degradation. dKV-Cache, in final, achieves from 2-10× speedup in inference, largely narrowing the gap between ARs and DLMs. We evaluate our dKV-Cache on several benchmarks, delivering acceleration across general language understanding, mathematical, and code-generation benchmarks. Experiments demonstrate that cache can also be used in DLMs, even in a training-free manner from current DLMs. The code is available at https://github.com/horseee/dKV-Cache.

## 1   Introduction

Diffusion language models (DLMs) [3, 25] have recently emerged as an alternative paradigm for text generation, inspired by the success of diffusion models in continuous domains like images [15, 48] and videos [28, 5]. Unlike autoregressive transformers (ARs)  [14, 43, 12] that generate text left-to-right at a time, a diffusion language model produces text by gradually refining a sequence of initially noisy tokens or masked tokens into a coherent output [24, 29]. Recent advancements in diffusion-based language models have underscored their versatility across both continuous [21] and discrete formulations [40, 3]. In particular, discrete diffusion models have shown competitive performance in language modeling [37], attracting growing interest for their potential to achieve faster decoding than traditional ARs while maintaining comparable generation quality.

One notable advantage of diffusion language models is their potential to decode an arbitrary number of tokens in parallel [49], whereas ARs require one forward pass per generated token. This parallel decoding paradigm offers the potential for improved wall-clock inference time and higher throughput. However, despite their theoretical efficiency, current DLMs remain substantially slower than ARs in practice. This inefficiency is primarily due to two factors: the incompatibility of DLMs with the

---

[*]Corresponding author

39th Conference on Neural Information Processing Systems (NeurIPS 2025).

KV-Cache mechanism [33, 13], and the large number of network evaluations for denoising [17]. Specifically, generating a sequence of length $L$ typically entails $L$ denoising steps, each involving a full bidirectional attention pass, leading to a cubic time complexity of $\mathcal{O}(L^3)$. In contrast, AR models utilize KV-Cache to reduce per-step complexity to $\mathcal{O}(L^2)$, achieving much faster inference overall.

In this paper, we tackle the challenge of integrating the KV-Cache mechanism into diffusion language models and operate without autoregressive or semi-autoregressive structures [2]. We identify two core reasons that prevent the direct usage of KV-Cache in DLMs. (1) KV-Cache hinges on the assumption that the key and value states of previously generated tokens remain fixed during subsequent decoding steps. This property is preserved in autoregressive models through the use of a causal attention mask, which restricts each token to attend only to earlier positions [11]. However, DLMs adopt a bidirectional attention mechanism, similar to non-autoregressive models [20], allowing every token to attend to all others in the sequence and making it nontrivial to reuse previously cached keys and values. (2) KV-Cache presupposes a fixed and left-to-right sequential decoding, where the position of the next token is deterministic. This assumption enables ARs to compute QKV states selectively only at the current decoding position. However, DLMs break this paradigm by supporting flexible generation orders. At each denoising step, any token position may be selected for update, which is a key advantage of DLMs for tasks involving long-range dependencies and holistic planning [51].

To solve this problem, we propose the **d**elayed KV-Cache, dKV-Cache, the KV-Cache for **d**iffusion language models. The core design of our proposed dKV-Cache centers on how to enable caching of key and value states across denoising steps in diffusion language models. A key insight motivating this design is that, although DLMs employ bidirectional attention, intuitively incompatible with caching, the representations of key and value are not fundamentally unreusable, but rather require delayed and conditioned reuse. In particular, we observe that the evolution of key and value states is strongly influenced by whether a token has been decoded. This behavior motivates a delayed caching strategy, wherein only the key and value states of decoded tokens would be cached, delayed from the ARs that caching occurs immediately upon input. We further propose a one-step delayed caching mechanism, in which the caching of key/value states is postponed by another denoising step. This intentional delay substantially boosts the performance and also reduces memory overhead for KV storage. Besides the delayed caching, we further propose a more aggressive decoding strategy to reduce the computational complexity of diffusion language models from $\mathcal{O}(L^3)$ to $\mathcal{O}(L^2)$ by restricting the caching to a compact subset of tokens: the delayed tokens and the current to-be-decoded tokens, and the window tokens.

Our experiments demonstrate that the proposed method achieves 2–10× speedup on existing 7B-scale diffusion language models, including LLaDA [37] and Dream [52], across a broad range of benchmarks such as general language understanding, code generation, and mathematical problem solving. These efficiency gains come with only minor and often negligible performance degradation, highlighting the practical value of our approach for accelerating DLM inference without training. Furthermore, we demonstrate that dKV-Cache is robust across variations in prefill length, output length, and the number of sampling steps.

**Contributions.** (1) We propose the first KV-Cache mechanism for diffusion language models by leveraging the evolving dynamics of token representations. We introduce a delay caching strategy compatible with bidirectional attention. (2) We propose two practical variants of our method: dKV-Cache-Decode, which enables long-term cache reuse, and dKV-Cache-Greedy, which reduces the per-step time complexity for faster decoding. (3) Extensive experiments on DLMs demonstrate that our approach achieves 2–10× inference speedup with minimal or negligible performance loss.

## 2 Related Work

### 2.1 Diffusion Language Models

Diffusion models [24, 41] model the data generation as the inversion of a forward-noise process and demonstrated impressive generation quality in image [38], video [5], and audio generation [16]. For diffusion models on language generation, [25, 3] extends DDPM to categorical data and defines the transition matrices for the corruption and denoising. [29] introduces the continuous-time diffusion over the continuous word-embedding space, and [21] closes the quality of generation on par with GPT-2 via a simplex defined over the vocabulary. Besides the diffusion in the continuous space [47, 53]

for discrete distribution, another approach seeks the path in discrete language diffusion models. [22] training BERT to learn the reverse process of a discrete diffusion process with an absorbing state in D3PM. SEDD [33] introduces score entropy training, a novel loss extending score-matching to discrete data and MDLM [40] shows that simple masked discrete diffusion is competitive to all previous kinds of DLMs. Block diffusion [2] extends the current non-autogressive [20] discrete language diffusion models into a semi-autoregressive one [21, 55], making it feasible to generate sequences of arbitrary length. [37, 19] scaling the masked diffusion language models to billions of parameters, achieving performance comparable to leading autoregressive LLMs.

## 2.2 Cache in Generative Models

Cache [45] is a small, fast memory that stores frequently accessed data, reducing the time that the CPU needs to fetch data from slower memory. Cache is first introduced in deep neural networks in transformers [44], where the KV-Cache caches previous tokens' key and value tensors. KV-Cache becomes a fundamental technique in transformers, and several improved techniques are proposed [18, 32, 26] to reduce the memory consumption of KV-Cache for long-context generation. Besides this, caches are also been explored in diffusion models for image [36, 46]. Those work leverage the temporal similarities between high-level features [35, 9], attention map [54, 31] to achieve faster inference of diffusion generation. This also has been explored in 3D generative modeling [50] and video generation [56, 34]. However, the cache for diffusion language models is less explored, especially the KV-Cache for diffusion language models. [2] explores the kv-cache in semi-autoregressive diffusion language models. This requires considering the KV-Cache in training, making twice the forward computation in the training and its form is still constrained in the autoregressive formula. [40] also considers cache, but under a strict condition that no new tokens have been calculated.

## 3 Methods

### 3.1 Preliminary

We primarily focus on continuous-time discrete language models, with particular attention to masked diffusion language models [40, 37], which have shown strong scalability to billion-parameter scales with high generation quality.

Consider a text sequence with $L$ tokens $\mathbf{x}_0^{1:L}$, sampled from the target distribution $p_{data}(\mathbf{x}_0)$. Each token is represented by a one-hot vector with $V$ categories, where $V$ is the vocabulary size. The forward process adds noise in the original sequence $\mathbf{x}_0$, which, in the discrete diffusion models here, takes the form of masking some of the tokens randomly. The masking process can be controlled by a transition matrix $\boldsymbol{U}_t$, where each element in this matrix $[\boldsymbol{U}_t]_{ij}$ represents the probability transition from token $i$ to token $j$ at step $t$ [3]. The denoising process is discretized into $T$ steps, and we define the continuous timestep $c(t) = t/T$, where $t \in \{0, 1, \ldots, T\}$. We use *timestep* in the continuous temporal space of the denoising process and *step* in the discrete space. The forward diffusion can be modelled as:

$$q\left(\boldsymbol{x}_{c(t)} \mid \boldsymbol{x}_0\right) = \mathrm{Cat}\left(\boldsymbol{x}_{c(t)}; \boldsymbol{p} = \boldsymbol{x}_0\overline{\boldsymbol{U}}_t\right), \quad \text{where} \quad \overline{\boldsymbol{U}}_t = \prod_{i=1}^{t} \boldsymbol{U}_i$$

Then we can get the corrupted $\mathbf{x}_0$. The absorbing form for the transition matrix is used here, where each token either remains the same or transfers to the special [MASK] token at the probability of $\beta_t$. The cumulated transition matrix $\overline{\boldsymbol{U}}_t$, as defined in the masked diffusion models, can be formulated as:

$$\left[\overline{\boldsymbol{U}}_t\right]_{ij} = \begin{cases} 1 & \text{if} \quad i = j = [\text{MASK}] \\ \bar{\alpha}_t & \text{if} \quad i = j \neq [\text{MASK}] \\ 1 - \bar{\alpha}_t & \text{if} \quad j = m, i \neq [\text{MASK}] \end{cases} \quad \text{with} \quad \bar{\alpha}_t = \prod_{i=1}^{t}(1 - \beta_i)$$

where $\bar{\alpha}_t$ linearly decrease to 0 as $t$ approaches T. The reverse process is learned by the models $\theta$, where $p_\theta(\mathbf{x}_{c(t-1)}|\mathbf{x}_{c(t)})$ is learned to approximate $q(\mathbf{x}_{c(t-1)}|\mathbf{x}_{c(t)}, \mathbf{x}_0)$. In $p_\theta(\mathbf{x}_{c(t-1)}|\mathbf{x}_{c(t)})$, the neural network with parameters $\theta$ is optimized to predict the clean tokens $\mathbf{x}_0$ given $\mathbf{x}_{c(t)}$.

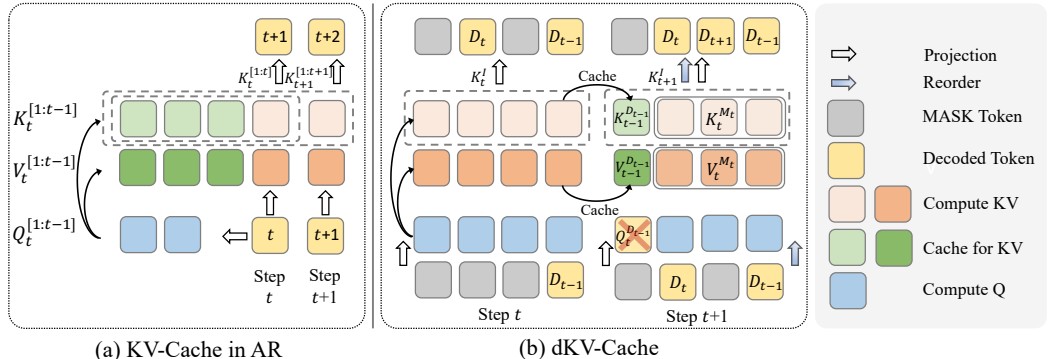

(a) KV-Cache in AR  (b) dKV-Cache

Figure 1: Illustration of dKV-Cache. At step $t$, no prior cache would be activated though the token $D_{t-1}$ has been decoded. $\mathbf{K}$ and $\mathbf{V}$ are delayed to the next step to be reordered and reused.

**Sampling Process of DLMs.** Given the noisy sequence $x_1^{1:L}$, which is consisted only with the masked token. Each timestep, the denoising model $p_\theta(\mathbf{x}_{c(t-1)}|\mathbf{x}_{c(t)})$ would be called, with $x_0$ predict first and then remasking the rest by $q(\mathbf{x}_{c(t-1)}|\mathbf{x}_0,\mathbf{x}_{c(t)})$. The unmasked tokens would remain unchanged during the later denoising process. Several strategies are used in the remasking stage, e.g., random remasking [3], keeping the most confident ones [7] or by selecting the topk positions with the largest margin between the two most probable values [27].

**Formulation of KV-Cache.** Since diffusion language models still use the transformer architecture (with or without GQA [1] would not affect our method), we simply grab the formulation of KV-Cache in ARs first. In a transformer decoder, each layer would project the current hidden states $\mathbf{h}_t$ into a query-key-value triplet $(\mathbf{Q}_t, \mathbf{K}_t, \mathbf{V}_t)$ via learned projection $\mathbf{W}_\mathbf{Q}, \mathbf{W}_\mathbf{K}$ and $\mathbf{W}_\mathbf{V}$. At step $t$, only the hidden states of the $t$-th tokens $\mathbf{h}_t^{[t]}$ would be calculated. The recursive KV-Cache update is appending the new key–value pair to the running buffered KV-Cache:

$$\mathbf{z}_t = \mathrm{softmax}\left(\frac{\mathbf{Q}_t^{[t]}\left(\mathbf{K}_t^{[1:t]}\right)^\top}{\sqrt{d_k}}\right)\mathbf{V}^{[1:t]} \quad \text{with} \quad \begin{cases} \mathbf{K}_t^{[1:t]} = \mathrm{concat}\left(\mathbf{K}_{t-1}^{[1:t-1]}, \mathbf{K}_t^{[t]}\right) \\ \mathbf{V}_t^{[1:t]} = \mathrm{concat}\left(\mathbf{V}_{t-1}^{[1:t-1]}, \mathbf{V}_t^{[t]}\right) \end{cases} \quad (1)$$

where $\mathbf{z}_t$ is the output of the attention head at step $t$ and the dot products are scaled down by $\sqrt{d_k}$.

## 3.2 Why KV-Cache Cannot be Used in DLMs?

The effectiveness of KV-Cache can be attributed to the reuse of previously computed K and V states, and the targeted computation only for the current decoding token. We conclude that standard KV-Cache is fundamentally incompatible with diffusion language models for two reasons:

- **Timestep-variant key and value states.** In the autoregressive setting, every time step shares a single, causally growing set of key and value tensors. $\mathbf{K}_m^{[1:t-1]}$ and $\mathbf{V}_m^{[1:t-1]}$ are the same at each step $m$ from $t-1$ and later with the help of causal attention mask. By contrast, DLMs employ a bidirectional attention mask; consequently, the key and value representations that each token can attend to at timestep $m$, $\mathbf{K}_m^{[1:t-1]}$ and $\mathbf{V}_m^{[1:t-1]}$, differ from those at timestep $n$. Put differently, $\mathbf{K}_m^{[1:t-1]} \neq \mathbf{K}_n^{[1:t-1]}$ and $\mathbf{V}_m^{[1:t-1]} \neq \mathbf{V}_n^{[1:t-1]}$ if $n \neq m$. The bidirectional attention introduced by diffusion language models therefore breaks the global reuse assumption that supports conventional KV-Cache.

- **Non-sequential decoding order.** Generation in DLMs does not follow a strictly left-to-right order. Instead, the model dynamically fills masked positions based on probabilities computed at each denoising step. As a result, the positions of decoded tokens are only revealed after the model forward pass, and the subsequent update may target any position in the sequence, rather than progressing sequentially. This uncertainty prevents us from pre-determining which token $i$ will require the computation of its hidden states $\mathbf{h}^{[i]}$ and its $\mathbf{Q}^{[i]}, \mathbf{K}^{[i]}$ and $\mathbf{V}^{[i]}$.

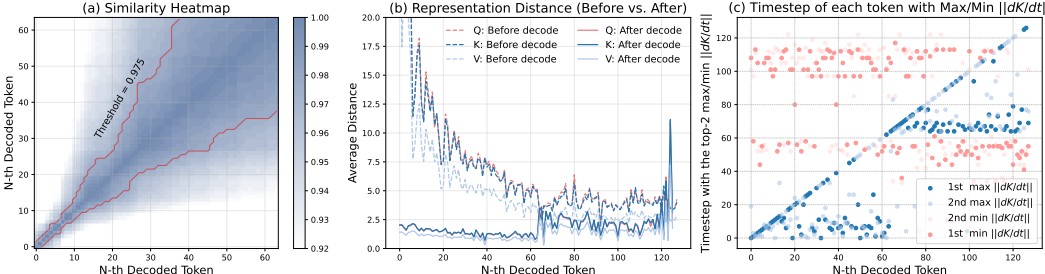

Figure 2: (a) We present a heatmap illustrating the pairwise similarities among the key states across different timesteps. Here we use LLaDA with $L = 128$, $T = 128$ and the block size is set to 64. We then compute the Euclidean distance between consecutive steps $t$ and $t + 1$ to analyze the dynamics of intermediate representations. (b) We report the average distance measured before and after the decoding of each token. (c) We highlight the top-2 steps exhibiting the largest and smallest changes in the key and value states for each token in their decoded order.

**Representation Dynamics for tokens in Diffusion Sampling** We investigate in DLMs whether $\mathbf{K}$ and $\mathbf{V}$ can be reused. We focus on the dynamics of $\mathbf{K}$ and $\mathbf{V}$ for each token, and the results are shown in Figure 2. Interestingly, we observe several noteworthy patterns in the dynamics of QKV states: (1) Despite step-to-step differences, the key and value embeddings, $\mathbf{K}$ and $\mathbf{V}$, exhibit consistently high similarity across timesteps, as shown in Figure 2(a). (2) Once a token is decoded, its representation becomes relatively stable in subsequent steps, whereas the representations of still-masked tokens continue to fluctuate significantly. This phenomenon is evident in Figure 2(b), where QKV fluctuations are more pronounced prior to decoding than thereafter. (3) The most substantial changes in $\mathbf{K}$ and $\mathbf{V}$ occur at the decoding step of each token and then in the early stages of the denoising process. This is reflected by prominent changes along the diagonal in Figure 2(c), corresponding to the $i$-th decoding step which decodes the $i$-th token.

These observations provide key insights into the temporal structure of discrete diffusion models and motivate the design of our KV-Cache mechanism for diffusion language modeling.

### 3.3 Delayed KV-Cache for Masked Diffusion Language Models

We first present a more general non-sequential KV-Cache formulation that replaces the contiguous slice $\mathrm{K}^{[1:t-1]}$ used in Eq.1 with an arbitrary-order index set $\mathcal{S}_t \subseteq \mathcal{I} = \{1, \dots, L\}$ and the next token position from the fixed $t$ to $D_t$. The cached keys and values gathered from previous steps are now $\mathbf{K}_t^{\mathcal{S}_t}$, which retrieves cached states at the positions specified by indexes in $\mathcal{S}_t$:

$$\mathbf{z}_t = \mathrm{softmax} \left( \frac{\mathbf{Q}_t^{D_t} \left( \mathbf{K}_t^{\mathcal{S}_t \cup \{D_t\}} \right)^\top}{\sqrt{d_k}} \right) \mathbf{V}_t^{\mathcal{S}_t \cup \{D_t\}} \ \mathrm{with} \ \begin{cases} \mathbf{K}_t^{\mathcal{S}_t \cup \{t\}} = \mathrm{concat\_reorder} \left( \mathbf{K}_{t-1}^{\mathcal{S}_t}, \mathbf{K}_t^{D_t} \right) \\ \mathbf{V}_t^{\mathcal{S}_t \cup \{t\}} = \mathrm{concat\_reorder} \left( \mathbf{V}_{t-1}^{\mathcal{S}_t}, \mathbf{V}_t^{D_t} \right) \end{cases}$$

(2)

where $D_t$ is the denoised token at step $t$. If we use random sampling for diffusion, then before any inference, we can know the decoding order of the sequence. We here use the notation of $D_t$, and later would not depend on knowing the decoding order. The operator $\mathrm{concat\_reorder}$ is proposed to enhance the efficiency of the indexing and gathering of K and V here. We explain in the appendix about this operator and how it works with ROPE [42] and how it accelerates.

We first extend the formulation in Eq.2 by generalizing the query input $\mathbf{Q}_t^{D_t}$ from the single-token decoding to the multiple and arbitrary token decoding. Specifically, at decoding step $t$, we construct a dynamic set $\mathcal{M}_t$, where $\mathcal{M}_t$ denotes the subset of tokens that are not yet finalized during generation. And with the $\mathbf{h}_t^{\mathcal{M}_t}$ calcuated, we also can get the corresponding $\mathbf{Q}_t^{\mathcal{M}_t}$, $\mathbf{K}_t^{\mathcal{M}_t}$ and $\mathbf{V}_t^{\mathcal{M}_t}$. We delay the caching of each token, not the time it is appended in the input, but rather on the step it is decoded:

$$\mathrm{concat\_reorder} ( \underbrace{\mathbf{K}_{t-1}^{\mathcal{S}_t}}_{\text{Cache from } t-1}, \underbrace{\mathbf{K}_t^{D_t}}_{\text{Calcualte at } t} ) \Rightarrow \mathrm{concat\_reorder} ( \underbrace{\mathbf{K}_{t-1}^{\mathcal{I} \setminus \mathcal{M}_t}}_{\text{Cache from } t-1}, \underbrace{\mathbf{K}_t^{\mathcal{M}_t}}_{\text{Calculate at } t} )$$

(3)

where $\mathcal{I}$ denotes the set of all tokens involved in the denoising process. This design reflects our core observation in the previous section: only the decoded tokens are eligible for caching, while

the remaining masked tokens must be re-encoded at each step. Besides, this method solves the problem that we need to predefine or predict the denoising order, as $\mathcal{M}_t$ is explicitly known at each step. Cached keys and values corresponding to $\mathcal{I} \setminus \mathcal{M}_t$ are reused across steps, while non-finalized positions are recomputed at each step to ensure correctness under bidirectional attention masking.

**One-step Delayed Caching.** As our analysis in Figure 2(c), the most significant change in $\mathbf{K}$ and $\mathbf{V}$ occurs exactly at the step where a token transitions from [MASK] to its decoded form. Currently, $\mathbf{K}_t^{D_t}$ would be used for caching, as it is no longer in the masked set $\mathcal{M}_t$. However, $\mathbf{K}_t^{D_t}$ can differ substantially from $\mathbf{K}_{t+1}^{D_t}$, and prematurely reusing it lead to severe performance degradation. To address this, we introduce one-step delayed caching. At timestep $t$, we use the masking state from the previous step, $\mathcal{M}_{t-1}$, to determine which tokens are cacheable. The method, named dKV-Cache-Decode, is finally formalized as:

$$\mathbf{z}_t = \text{softmax}\left( \frac{\mathbf{Q}_t^{\mathcal{M}_{t-1}} \left( \mathbf{K}_t^{\mathcal{I}} \right)^\top}{\sqrt{d_k}} \right) \mathbf{V}_t^{\mathcal{I}} \text{ with } \begin{cases} \mathbf{K}_t^{\mathcal{I}} = \text{concat\_reorder}\left( \mathbf{K}_{t-1}^{\mathcal{I} \setminus \mathcal{M}_{t-1}}, \mathbf{K}_t^{\mathcal{M}_{t-1}} \right) \\ \mathbf{V}_t^{\mathcal{I}} = \text{concat\_reorder}\left( \mathbf{V}_{t-1}^{\mathcal{I} \setminus \mathcal{M}_{t-1}}, \mathbf{V}_t^{\mathcal{M}_{t-1}} \right) \end{cases}$$
(4)

While this slightly reduces efficiency, we find it to be critical for maintaining accuracy and stability in the proposed dKV-Cache mechanism for diffusion language models.

**Cache Refreshing Mechanism.** While it is possible to apply caching after each token is decoded and reuse it throughout the denoising process, in practice, when the sequence is sufficiently long, occasionally recomputing the cache incurs small computational overhead. To maintain consistency and improve correctness during decoding, we add a cache refreshing mechanism. Every N steps, the stored cache would be discarded and refreshed. The calculation would revert back to the normal calculation, resulting in an empty set $\emptyset$ to replace $\mathcal{M}_{t-1}$ for this refresh step in Eq.4.

**dKV-Cache-Prefill and dKV-Cache-PD.** The set $\mathcal{M}_{t-1}$ can be further divided into two subsets: decoded tokens and always-decoded tokens, i.e., prefill tokens. Our experiments show that prefill tokens primarily attend to each other, indicating limited influence from later tokens. Based on this, we adopt a special strategy, dKV-Cache-Prefill, that caches prefill tokens without refreshing. This design aligns with the disaggregation of prefill and decoding phases [57] for serving DLMs. Building on this, we have another variant, dKV-Cache-PD, which intermittently refreshes only the newly decoded tokens, while keeping the key and values of prefill tokens without any recomputation.

### 3.4 dKV-Cache-Greedy: Greedy Formulation of dKV-Cache

However, the above method still incurs $\mathcal{O}(L^3)$ complexity, which is less efficient than the $\mathcal{O}(L^2)$ complexity of ARs. This is primarily because $\mathcal{M}_t$ initially consists of the entire sequence with $L$ tokens and only narrows to a single token at the end. To improve efficiency, it is essential to decouple $|\mathcal{M}_t|$ for each step from the sequence length $L$.

Instead of the minimally refreshed caches proposed above, we adopt a more relaxed cache mechanism to refresh more so as to mitigate the performance degradation caused by stale $\mathbf{K}$ and $\mathbf{V}$. Building on our earlier observation that token representations undergo significant changes at their decoding step, we define $\mathcal{M}_t$ to include only three components: the token at the current step $D_t$, the token from the previous step $D(t-1)$ (motivated by one-step delayed caching), and a local window $\mathcal{W}(t)$. For this local window, we extend it to include the token itself and its neighboring tokens within a fixed-size window $\mathcal{W}_t = \left\{ x_i \mid i \in \left[ D_t - \left\lceil \frac{w}{2} \right\rceil, D_t + \left\lfloor \frac{w}{2} \right\rfloor \right] \right\}$, where $w$ is the window size. We evaluated local windows centered at both $D_t$ and $D_{t-1}$, and found that the latter yields better performance. Since the window size $|\mathcal{W}_t|$ is fixed (set to at most 6 in our experiments), this strategy introduces additional computation but retains an overall time complexity of $\mathcal{O}(L^2)$.

## 4 Experiments

### 4.1 Experimental Setup

We tested our method under the original evaluation benchmark of LLaDA [37] and Dream [52]. **Datasets:** We conduct comprehensive evaluations across a diverse set of benchmarks that assess

Table 1: Benchmark results on LLaDA-8B-Instruct. We use zero-shot evaluation here. Detailed configuration is listed in the Appendix. We set the cache refresh step for dKV-Cache-Decode to be 8 and dKV-Cache-Greedy to be 2. The window size of dKV-Cache-Greedy is listed in the bracket.

| Remasking | Base (random) | Few-Steps (random) | dKV-Cache-Greedy (random) | dKV-Cache-Greedy w/ Window (random) | Base (confidence) | Half-Steps (confidence) | dKV-Cache-Decode (confidence) |
|---|---|---|---|---|---|---|---|
| MMLU | 51.79 | 43.19 | 45.77 | 47.70 (4) | 51.11 | 51.11 | 51.00 |
| | 30.20 | 47.49 (1.67×) | 50.56 (1.57×) | 45.72 (1.51×) | 28.27 | 55.80 (1.97×) | 66.52 (2.35×) |
| GSM8K | 72.25 | 65.58 | 67.93 | 68.23 (4) | 77.56 | 77.91 | 78.85 |
| | 15.16 | 24.08 (1.59×) | 25.47 (1.68×) | 24.76 (1.63×) | 14.31 | 28.71 (2.00×) | 27.50 (1.92×) |
| Math500 | 27.4 | 21.8 | 26.0 | 27.0 (4) | 36.6 | 34.2 | 36.8 |
| | 12.00 | 19.36 (1.61×) | 20.34 (1.70×) | 19.86 (1.66×) | 11.53 | 23.10 (2.00×) | 24.46 (2.12×) |
| GPQA | 27.46 | 24.78 | 26.79 | 28.35 (4) | 30.80 | 27.68 | 28.13 |
| | 11.40 | 18.59 (1.63×) | 19.27 (1.69×) | 18.26 (1.60×) | 11.86 | 23.88 (2.01×) | 28.73 (2.42×) |
| HumanEval | 19.88 | 15.61 | 15.13 | 15.37 (4) | 39.63 | 33.54 | 46.34 |
| | 7.50 | 12.50 (1.67×) | 12.31 (1.64×) | 12.13 (1.62×) | 7.08 | 14.18 (2.00×) | 13.76 (1.83×) |
| MBPP | 21.4 | 15.6 | 17.8 | 20.4 (2) | 40.4 | 33.8 | 40.4 |
| | 7.51 | 12.97 (1.73×) | 12.55 (1.67×) | 12.44 (1.66×) | 7.50 | 15.01 (2.00×) | 13.93 (1.86×) |

Table 2: Benchmark results on Dream-Base-7B. We use the few-shot ICL here and the configuration is in Appendix. We set the cache refresh interval for dKV-Cache-Decode and dKV-Cache-PD to 4.

| | | Dream-7B | Half-Steps | dKV-Cache-Decode | dKV-Cache-Prefill | dKV-Cache-PD |
|---|---|---|---|---|---|---|
| GSM8K (8-shot) L = 256 | T = 256 | 76.88 | 68.08 | 76.57 | 75.66 | 74.07 |
| | | 15.1 (1.00×) | 30.3 (2.00×) | 31.6 (2.09×) | 53.6 (3.55×) | 50.2 (3.32×) |
| | T = 128 | 68.81 | 46.63 | 65.35 | 65.96 | 63.31 |
| | | 30.3 (2.01×) | 60.5 (4.01×) | 62.31 (4.13×) | 107.4 (7.11×) | 99.5 (6.6×) |
| MBPP (3-shot) L = 512 | T = 512 | 55.8 | 45.2 | 53.4 | 55.2 | 51.0 |
| | | 5.4 (1.00×) | 10.8 (2.00×) | 10.4 (1.93×) | 13.6 (2.52×) | 14.5 (2.69×) |
| | T = 256 | 45.2 | 26.2 | 43.4 | 41.8 | 42.6 |
| | | 10.8 (2.00×) | 21.5 (3.98×) | 20.6 (3.81×) | 27.1 (5.02×) | 28.9 (5.35 ×) |
| HumanEval (0-shot) L = 512 | T = 512 | 57.93 | 37.20 | 57.32 | 56.10 | 59.76 |
| | | 10.3 (1.00×) | 20.5 (1.99×) | 15.5 (1.50×) | 14.4 (1.40×) | 17.4 (1.69×) |
| | T = 256 | 37.20 | 18.29 | 31.70 | 33.54 | 31.70 |
| | | 20.5 (1.99×) | 40.9 (3.97×) | 31.1 (3.02×) | 28.7 (2.79×) | 34.8 (3.38×) |

general language understanding [23], mathematical reasoning [10, 30, 39], and code generation [8, 4]. Since the multi-choice evaluation based on the token likelihood doesn't require more than one step for inference, we request the models to generate the answer letter and match the generated answer with the ground-truth answer. **Evaluation:** We follow the prompt in simple-evals[2] for LLaDA, making the model reason step by step. On Dream, we follow the evaluation setting of Dream to conduct few-shot in-context learning [6][3]. Other implementation details are listed in the Appendix. **Baseline:** We choose the few-step sampling method (50% steps for Half-Steps and 62.5% steps for Few-Steps) as our baseline and select the number of steps such that their sampling speed is comparable to or slower than ours, and showing that our method can have better performance.

**Metric.** We report the accuracy for performance and token/s for speed. We tested the speed on A6000 (for LLaDA) and H20 (for Dream). Besides this, we use one more metric to show the cache ratio, calculated as: $\frac{1}{T} \sum_{i=1}^{T} |\mathcal{T}_i^{\text{cache}}| / |\mathcal{T}_i|$, where $T$ is the total number of decoding steps. $\mathcal{T}_i$ denotes the number of tokens processed at step $i$ (normally the whole sequence) and $\mathcal{T}_i^{\text{cache}}$ is the subset of tokens whose KV pairs are reused from cache at timestep $i$, which is $|\mathcal{M}_i|$.

## 4.2 Performance and Speed with dKV-Cache

We begin by addressing the central question: What are the performance trade-offs and speedups introduced by applying dKV-Cache? For LLaDA, we evaluate two variants, dKV-Cache-Greedy and dKV-Cache-Decode, against baselines that accelerate generation by halving or reducing the number of denoising steps. As shown in Table 1, dKV-Cache-Greedy consistently outperforms few-step

---

[2]https://github.com/openai/simple-evals
[3]https://github.com/EleutherAI/lm-evaluation-harness

Table 3: Results on long prefill settings. dKV-Cache-Decode uses a refresh step of 4; dKV-Cache-Prefill never refreshes.

| | | Dream-7B | Half-Steps | dKV-Cache-Decode | dKV-Cache-Prefill |
|---|---|---|---|---|---|
| MMLU (5-shot) L = 8 | T = 8 | 72.19 
 9.1 (1.00×) | 72.21 
 18.1 (1.99×) | 71.74 
 25.2 (2.77×) | 71.76 
 57.6 (6.33×) |
| | T = 4 | 72.21 
 18.1 (1.99×) | 71.63 
 36.1 (3.97×) | 71.69 
 49.2 (5.41×) | 71.71 
 67.3 (7.40×) |
| GPQA (5-shot) L = 128 | T = 128 | 36.83 
 7.4 (1.00×) | 35.49 
 14.7 (1.99×) | 35.71 
 18.2 (2.46×) | 35.27 
 75.40 (10.19×) |
| | T = 64 | 35.49 
 14.7 (1.99×) | 35.27 
 29.4 (3.97×) | 34.15 
 36.8 (4.97×) | 35.27 
 139.9 (18.91×) |

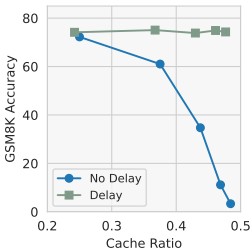

Figure 3: Effect of one-step delayed caching

baselines across most benchmarks, except for HumanEval. Notably, integrating a lightweight cache window yields substantial gains with negligible computational overhead. For dKV-Cache-Decode, **dKV-Cache-Decode achieves near-lossless performance with a high cache ratio and only a few refresh steps**. Among all strategies, dKV-Cache-Decode delivers the best trade-off, outperforming both dKV-Cache-Greedy and the baselines. Crucially, it maintains accuracy nearly indistinguishable from the full model, demonstrating that KV-Cache can also be applied in diffusion language models without sacrificing performance. Since dKV-Cache-Greedy relies on a predefined (e.g., random) decoding order, which results in a marked accuracy drop relative to low-confidence remasking, we concentrate our experiments more on dKV-Cache-Decode. We provide several case studies in the Appendix that compare the generated text before and after applying dKV-Cache.

The results on Dream are shown in Table 2 and Table 3. There is a small difference in the position of the decoded token since Dream is adapted from auto-regressive models and shifts the token position. We provide a detailed illustration in the appendix for this. Due to the use of few-shot in-context learning, the model requires a long input context, leading to significant overhead from encoding those tokens repeatedly. In this setting, dKV-Cache-Prefill provides substantial speed improvements; for instance, on MMLU and GPQA, it achieves up to a 10× acceleration. Across all tested datasets, we observe that dKV-Cache largely outperforms the baseline under different prefilling and decoding lengths. We further evaluate the impact of applying dKV-Cache to few-step diffusion models and observe consistent trends: as the number of diffusion steps increases, our method yields even larger gains over the baseline. For example, on GSM8K with a decoding length of 256, the baseline model with 64 steps achieves 46.63 Pass@1 with a 4× speedup, whereas dKV-Cache attains a 6.6× speedup while significantly improving performance to 63.31 (+16.68).

### 4.3 Analysis

**The one-step delay in dKV-Cache.** Figure 3 illustrates the impact of applying a one-step delay to the cache mechanism. Without the delay, performance remains acceptable at low cache ratios but degrades rapidly as the cache ratio increases. Introducing a one-step delay stabilizes the generation quality, enabling the model to maintain nearly lossless performance even under high cache ratios.

**Performance on different decoding length, denoising steps and refreshing steps.** Figure 4 presents the results of applying dKV-Cache-Decode and dKV-Cache-Greedy under various configurations, including different decoding lengths, refresh intervals, sampling steps, and window sizes. Overall, our method consistently achieves performance comparable to the original model without dKV-Cache, effectively pushing the Pareto front forward. We observe the following findings: (1) Decoding robustness: The performance impact of our method is largely insensitive to the number of decoding steps, indicating strong robustness across varying generation lengths and sampling steps. (2) Enhanced long-form generation: In tasks involving longer outputs (e.g., L = 512), our method outperforms the baseline, even improving generation quality from 80.97% to 83.13% on GSM8K and from 39.63% to 46.34% for HumanEval. The results imply a potential inefficiency in how bidirectional attention aggregates contextual signals, pointing to redundancy or underutilization in long-context modeling. (3) Effectiveness with only a few refreshing: Even with infrequent refreshes (e.g., every 16 steps), the performance degradation remains small. (4) Effectiveness of local windows: Incorporating a local window notably enhances the performance of dKV-Cache-Greedy with minimal additional computational cost.

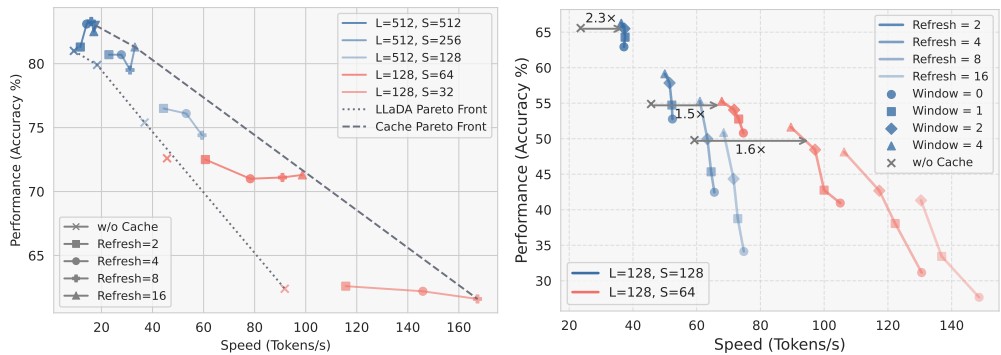

Figure 4: dKV-Cache-Decode(left) and dKV-Cache-Greedy(right) on GSM8K with different settings: decoding length $L$, sampling steps $S$, refresh intervals and the window size.

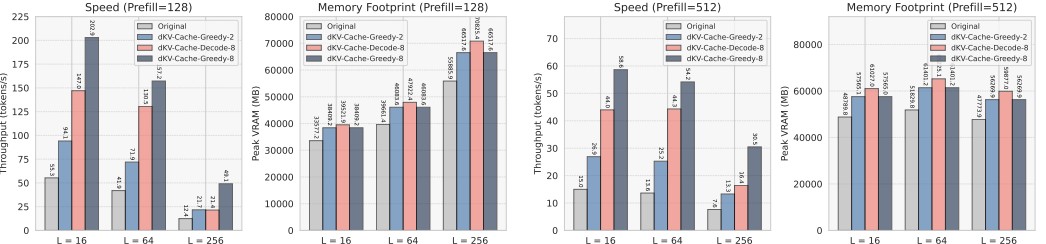

Figure 5: Speed and memory for dKV-Cache-Decode and dKV-Cache-Greedy. The number (2 and 8) means that in every n steps, the cache would be refreshed.

**Memory and speed analysis.** We analyze the speed and memory footprint of dKV-Cache-Decode and dKV-Cache-Greedy across varying decoding and prefill lengths. Our method achieves substantial inference acceleration, ranging from $1.75\times$ to $3.3\times$, while introducing only a modest increase in memory usage. Notably, dKV-Cache-Greedy demonstrates greater potential for accelerating inference while dKV-Cache-Decode would be capped. In our main experiments, we observed that setting the refresh interval larger than 2 for dKV-Cache-Greedy may degrade performance. However, under the same refresh interval, dKV-Cache-Greedy consistently achieves higher speedups than dKV-Cache-Decode, highlighting its potential advantage when refresh frequency is relaxed.

## 5 Conclusions and Limitations

In this work, we explore the feasibility of incorporating the caching mechanism into diffusion language models. Specifically, we propose a delayed KV-Cache for DLMs, motivated by our empirical observations on the dynamics of the token representations throughout the diffusion process. We introduce two cache variants, dKV-Cache-Decode and dKV-Cache-Greedy, each designed to leverage delayed caching for improved compatibility with diffusion-based generation. Our analysis reveals that introducing a delay is crucial for the cache to function effectively in this setting. Extensive experiments demonstrate that our approach largely accelerates inference while maintaining model performance. One primary limitation of this work lies in its focus on algorithmic design in isolation. While our proposed method introduces an effective caching mechanism from a purely algorithmic perspective, diffusion language models also exhibit substantial room for improvement at the system level. We believe that future research integrating algorithmic innovations with system-level optimization, such as memory management, parallelism, and hardware-aware execution, could unlock further efficiency gains and performance improvements for DLMs.

## Acknowledgment

This project is supported by the Ministry of Education, Singapore, under its Academic Research Fund Tier 2 (Award Number: MOE-T2EP20122-0006).

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

# A  Design for concat_reorder

concat_reorder is our implementation of dKV-Cache designed to improve the speed of dKV-Cache in diffusion language models. Unlike standard KV-Cache used in autoregressive models, dKV-Cache requires gathering and scattering keys and values from arbitrary positions, introducing indexing operations that are less efficient than the simple concatenation in contiguous space used in ARs.

In dKV-Cache, the cache process involves two additional indexing operations: (1) At the cache step: After computing keys and values, we need to gather the corresponding states of cached tokens at non-continuous positions. (2) At the reuse step: to obtain the whole matrices for key and value, we need to scatter these vectors back to their original positions in the sequence. In contrast, KV-Cache in ARs only requires matrix slicing and concatenation, making it significantly more efficient.

To minimize the overhead of gathering and scattering, we propose an algorithm similar to that of standard KV-Cache to avoid too many indexing operations. The key idea is to reorder token positions during the forward calculation of the Transformer, placing all cached tokens contiguously on one side (e.g., left) and newly decoded tokens on the other. This allows us to move parts of the indexing operation to the token level (matrices with shape [B, L]) instead of the intermediate states (matrices with shape [B, L, D]):

- At Step t-1: Gather the cached key $\mathbf{K}_{t-1}^{\mathcal{I}\setminus\mathcal{M}_{t-1}}$ and value states $\mathbf{V}_{t-1}^{\mathcal{I}\setminus\mathcal{M}_{t-1}}$ based on the position index $\mathcal{I}\setminus\mathcal{M}_{t-1}$ with one indexing operation.

- At Step t: Reorder the sequence, making the cached tokens (at position $\mathcal{I}\setminus\mathcal{M}_{t-1}$) at the left, and uncached tokens (at position $\mathcal{M}_{t-1}$) at the right.

- At Step t: Using concat_reorder for $\left(\mathbf{K}_{t-1}^{\mathcal{I}\setminus\mathcal{M}_{t-1}}, \mathbf{K}_t^{\mathcal{M}_{t-1}}\right)$ and for $\left(\mathbf{V}_{t-1}^{\mathcal{I}\setminus\mathcal{M}_{t-1}}, \mathbf{V}_t^{\mathcal{M}_{t-1}}\right)$: First, concatenate the cached and current key/value states directly without further gathering/scattering (**concat**, for getting all K and V to calculate attention), and reorder the whole KV matrics based on $\mathbf{V}_t^{\mathcal{I}\setminus\mathcal{M}_t}$ to get the cached states for the next step (**reorder**, for obtaining the cache).

The reorder operation is to know the position mapping from $[\mathcal{I}\setminus\mathcal{M}_{t-1}; \mathcal{M}_{t-1}]$ to $[\mathcal{I}\setminus\mathcal{M}_t; \mathcal{M}_t]$. For example, if the unmasked position at $t-1$ is [2, 4, 5] from a sequence of 8 tokens, and at step $t+1$ is [2, 4, 5, 7]. Then $[\mathcal{I}\setminus\mathcal{M}_{t-1}; \mathcal{M}_{t-1}]$ would be [2, 4, 5, 0, 1, 3, 6, 7], and $[\mathcal{I}\setminus\mathcal{M}_t; \mathcal{M}_t]$ would be [2, 4, 5, 7, 0, 1, 3, 6]. The mapping would be [0, 1, 2, 7, 3, 4, 5, 6], and we only need to get the corresponding entries [0, 1, 2, 7] from $\left[\mathbf{K}_{t-1}^{\mathcal{I}\setminus\mathcal{M}_{t-1}}; \mathbf{K}_t^{\mathcal{M}_{t-1}}\right]$ and $\left[\mathbf{V}_{t-1}^{\mathcal{I}\setminus\mathcal{M}_{t-1}}; \mathbf{V}_t^{\mathcal{M}_{t-1}}\right]$.

The only remaining thing is that the change of token position would impact the positional encoding. However, this is easy to solve; we can also reorder the positional embedding. Reordering positional embeddings is required only once per model evaluation and can be shared across layers, thus, it would not cost much time.

Furthermore, since our method introduces a one-step shift in caching, the position of cached tokens at step $t$ corresponds to the token positions decoded from step $t-1$. This alignment allows us to track which key and value entries need to be cached without storing the entire key/value matrices, which, to cache which tokens, can only be known after the decoding results at step $t$.

We present the pseudo-algorithm of our approach in Algorithm 1. While it largely improves inference speed over the naive implementation, the concat and reorder operations still introduce some overhead. We believe there is substantial potential for further optimization.

# B  Design for Dream

Dream has a different caching strategy with LLaDA. The main reason for this is that Dream is adapted from pre-trained autoregressive models, which would make the position of output align with the probability of the next token, instead of the current token in the traditional setting of masked diffusion models. This would make a difference in the caching strategy and we investigate between different designs of those caching strategies:

---

**Algorithm 1** Pseudo code for dKV-Cache-Decode. We take step t as an example.

---

**Require:** Sequence $\mathbf{x}_{c(t)}^{1:L}$ at step $t$ (Simplied as $\mathbf{x}$), position index of masked tokens $\mathcal{M}_t$, cached Key $\mathbf{K}_{t-1}^{\mathcal{I} \setminus \mathcal{M}_{t-1}}$ and Value $\mathbf{V}_{t-1}^{\mathcal{I} \setminus \mathcal{M}_{t-1}}$

1: $\mathbf{x}' \leftarrow \mathbf{x}[\mathcal{M}_{t-1}]$        ▷ $\mathcal{M}_{t-1}$: $t-1$ for one-step shift
2: $\mathbf{PE}' \leftarrow \big[\mathbf{PE}[\mathcal{I} \setminus \mathcal{M}_{t-1}]\,;\,\mathbf{PE}[\mathcal{M}_{t-1}]\big]$    ▷ Positional embeddings: cached on left, uncached on right
3: $\mathbf{Q}_t^{\mathcal{M}_t}, \mathbf{K}_t^{\mathcal{M}_t}, \mathbf{V}_t^{\mathcal{M}_t} \leftarrow \mathcal{T}(\mathbf{x}')$        ▷ $\mathcal{T}$: Calculation in Transformer to get Q, K and V
4: $\mathbf{K}_t^{\mathcal{I}} \leftarrow \text{Concat}\left(\mathbf{K}_{t-1}^{\mathcal{I} \setminus \mathcal{M}_{t-1}}, \mathbf{K}_t^{\mathcal{M}_{t-1}}\right), \mathbf{V}_t^{\mathcal{I}} \leftarrow \text{Concat}\left(\mathbf{V}_{t-1}^{\mathcal{I} \setminus \mathcal{M}_{t-1}}, \mathbf{V}_t^{\mathcal{M}_{t-1}}\right)$    ▷ Get all K and V
5: $\mathbf{K}_t^{\mathcal{I} \setminus \mathcal{M}_t} \leftarrow \text{Reorder}(\mathbf{K}_t^{\mathcal{I}}, I'), \mathbf{V}_t^{\mathcal{I} \setminus \mathcal{M}_t} \leftarrow \text{Reorder}(\mathbf{V}_t^{\mathcal{I}}, I')$    ▷ $\mathcal{I}'$ : The index of $\mathcal{I} \setminus \mathcal{M}_t$ in the $\big[\mathbf{x}[\mathcal{I} \setminus \mathcal{M}_{t-1}]\,;\,\mathbf{x}[\mathcal{M}_{t-1}]\big]$
6: $p' \leftarrow \mathcal{A}(\mathbf{Q}_t^{\mathcal{M}_t}, \mathbf{K}_t^{\mathcal{I}}, \mathbf{V}_t^{\mathcal{I}})$
7: $p \leftarrow \text{Scatter}(p', \mathcal{M}_{t-1})$        ▷ Put the token logits back to the original position

8: **Return** $p, \mathbf{K}_t^{\mathcal{I} \setminus \mathcal{M}_t}, \mathbf{V}_t^{\mathcal{I} \setminus \mathcal{M}_t}$

---

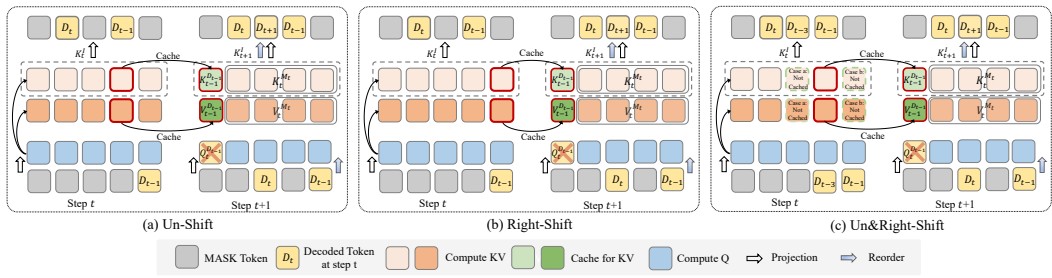

Figure 6: Three variants for the caching strategy for diffusion language models adapted from auto-regressive language models, which would have shifted output position.

- Un-Shift, Figure 6(a): We cache the unshifted token representations. Specifically, for the t-th token, we store its key and value as $\mathbf{K}^t$ and $\mathbf{V}^t$ at position $t$.

- Right-Shift, Figure 6(b): Given that the hidden state is highly sensitive to changes in input, we also explore a right-shifted variant. Here, for the t-th token, we cache $\mathbf{K}^{t+1}$ and $\mathbf{V}^{t+1}$.

- Un&Right-Sift, Figure 6(c): We introduce a stricter variant where caching is conditioned on both input stability and decoding completion. For the t-th token, we cache its features only after its input is fixed and it has been decoded.

The one-step shift is still used here. For example, in the right-shift variant, the t-th token is fed into the model at position t+1 in the next step, and we cache its output $\mathbf{K}^{t+1}$ and $\mathbf{V}^{t+1}$ then. The results are shown in Table 4, where Un&right-shift would have the best performance, and the right shift would largely harm the model performance. However, we use the Un-Shift in our main experiment, since Un&right-shift is incompatible with the above concat_reorder.

Table 4: Comparison between different types of caching strategy for Dream-Base-7B.

|  | Un-Shift | Right-Shift | Un&Right Shift |
|---|---|---|---|
| MMLU | 71.78 | 64.60 | 71.73 |
| GSM8K | 76.34 | 32.68 | 77.71 |

## C    Evaluation Details

### C.1    For LLaDA

We re-implemented the evaluation of LLaDA on those reported datasets. We generate and extract the final answer instead of comparing the log prob in the multiple-choice question. Thus, the result of MMLU and GPQA is lower than reported since the model sometimes cannot generate the answer in

the given format or does not generate the answer. We show the configuration of each experiment in Table 5

Table 5: Configurations of experiments on LLaDA-Instruct.

| Remasking | Base (random / confidence) Configuration | Few-Steps (random) Steps $T$ | dKV-Cache-Greedy (random) Cache Interval | dKV-Cache-Greedy (random) Window Size | Half-Steps (confidence) Steps $T$ | dKV-Cache-Decode (confidence) Cache Interval |
|---|---|---|---|---|---|---|
| MMLU | L=32, T=32, B=16 | T=20 | 2 | 4 | T=16 | 8 |
| GSM8K | L=256, T=256, B=32 | T=160 | 2 | 4 | T=128 | 8 |
| Math500 | L=256, T=256, B=64 | T=160 | 2 | 4 | T=128 | 8 |
| GPQA | L=128, T=128, B=64 | T=80 | 2 | 4 | T=64 | 8 |
| HumanEval | L=512, T=512, B=32 | T=320 | 2 | 4 | T=256 | 8 |
| MBPP | L=512, T=512, B=32 | T=320 | 2 | 2 | T=256 | 8 |

## C.2 For Dream

We follow the original evaluation pipeline for Dream[4] and we also adopt two datasets, MMLU and GPQA, to generate the answer instead of comparing the probabilities. We follow all the hyperparameters set in the evaluation script, including the temperature, the remasking strategy, top_p and the number of few-shot in in-context learning.

## D Impact of batch size on speed

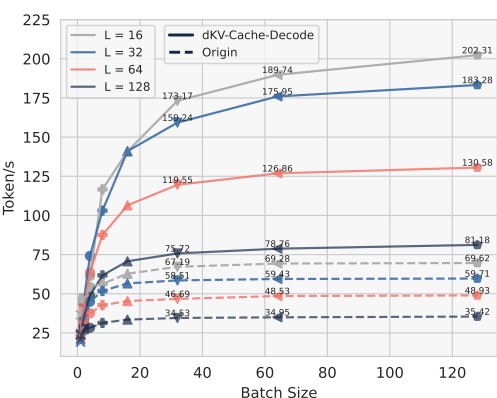

Figure 7: Impact of batch size on decoding speed. Evaluated on LLaDA with a single NVIDIA H20; prefill length fixed at 100 tokens.

Our inference pipeline relies heavily on indexing operations, gathers and scatters, that generate a stream of small, non-contiguous memory accesses. At a batch size of one, these uncoalesced reads make the inference memory-bound. As a result, the GPU's compute units sit idle waiting for data. In some cases with the batch size equal to 1, inference with caching can actually underperform the unaccelerated baseline. By contrast, scaling up the batch size can solve this problem and deliver far greater, more stable speed-up ratios over the baseline.

## E Case Study

We provide the case study of our method in Table 6 and Table 7 and Table 8.

---

[4]https://github.com/HKUNLP/Dream/blob/main/eval/eval_dream_gen.sh

Table 6: Case study on LLaDA-8B-Instruct. Question: Lily can run 12 kilometers per hour for 4 hours. After that, she runs 6 kilometers per hour. How many kilometers can she run in 8 hours?

| Model | Generate Text (L=128, T=128, B=64) |
|---|---|
| LLaDA-8B-Instruct (low confidence) | 1. Calculate the distance Lily runs in the first 4 hours:
- Speed: 12 kilometers per hour
- Time: 4 hours
- Distance: 12 km/h * 4 h = 48 km
2. Calculate the distance Lily runs in the next 4 hours:
- Speed: 6 kilometers per hour
- Time: 4 hours
- Distance: 6 km/h * 4 h = 24 km
3. Add the distances together:
- Total distance: \boxed{72} km |
| dKV-Cache-Decode, Refresh = 4 | 1. Calculate the distance Lily runs in the first 4 hours:
- Distance = Speed × Time = 12 km/h × 4 h = 48 km
2. Calculate the distance Lily runs in the next 4 hours:
- Distance = Speed × Time = 6 km/h × 4 h = 24 km
3. Add the distances from both periods to find the total distance Lily runs in 8 hours:
- Total Distance = 48 km + 24 km = 72 km
\boxed{72 km} |
| dKV-Cache-Decode, Refresh=16 | 1. Calculate the distance Lily runs in the first 4 hours:
12 km/hour * 4 hours = 48 km
2. Calculate the distance Lily runs in the next 4 hours:
6 km/hour * 4 hours = 24 km
3. Add the distances from the first 4 hours and the next 4 hours to find the total distance she runs:
48 km + 24 km = 72 km
Therefore, Lily can run \boxed{72} kilometers in 8 hours. |
| dKV-Cache-Greedy, Refresh = 2 | To determine how many kilometers Lily can run in 8 hours, we need to break down her running schedule and sum the distances.
First, calculate the distance she runs at 12 kilometers per hour for 4 hours:
$$12 * 4 = 48$$
After 4 hours, she runs 6 kilometers per hour for the remaining 4 hours:
$$6 * 4 = 24$$
Now, add the distances:
$$48 + 24 = 72$$
\boxed{72} |
| dKV-Cache-Greedy, Refresh = 4 (random) | For the first 4 hours Lily runs at 12 kilometers per hour, so she runs 12 * 4 = 48 kilometers. 8 hours - 4 hours are left after the first 4 (8 - 4 = 4 hours).
For the next 4 hours, she runs at 6 kilometers per hour, so she runs 6 * 4 = 24 kilometers.
Adding the two distances together, Lily can run 48 + 24 = 72 kilometers in 8 hours.
Answer: \boxed{72} |

Table 7: Case study on Dream-Instruct-7B. Request: Write a story that ends with "Finally, Joey and Rachel get married."

| Model | Generate Text (L=256, T=256) |
|---|---|
| Dream-Instruct-7B (Entropy) | Once upon a time, there was a young man named Joey who lived in a small town in the middle of nowhere. He was a kind and gentle soul who loved to sing and play the guitar. One day, he met a beautiful girl named Rachel who was also a talented singer and musician. They fell in love and decided to get married. However, there was a problem. Joey's parents were against the marriage because they didn't know Rachel well enough. They thought that she was not a good match for their son. Rachel, on the other hand, was not worried about Joey's parents' opinion. She knew that she loved Joey and that he was the one for her. She decided to try and win over Joey's parents by showing them how much she cared about him. She started by cooking for them and inviting them to her house for dinner. She also took them on a trip to the mountains where they could enjoy the scenery and the fresh air. After a while, Joey's parents started to see Rachel in a different light. They realized that she was a good person and that she loved Joey very much. They decided to give their blessing to the marriage. Finally, Joey and Rachel got married and lived happily ever after. |
| dKV-Cache-Decode, Refresh = 16 | Once upon a time, in a small town called New York, there lived a young man named Joey and a young woman named Rachel. They were both very happy with their lives, but they were also very different from each other. Joey was a very outgoing person who loved to be around people, while Rachel was a very introverted person who loved to be alone. One day, Joey and Rachel met at a coffee shop and fell in love. They spent hours talking and laughing together, and they realized that they were perfect for each other. They decided to be together and build their lives together. As the years went by, Joey and Rachel grew closer and closer. They shared their dreams and their fears, and they supported each other through thick and thin. They were a perfect match, and they knew that they were meant to be together. Finally, Joey and Rachel got married. They had a beautiful wedding with all their friends and family in attendance. They were so happy to be together, and they knew that they would always be together. |
| dKV-Cache-Prefill | Once upon a time, in a small town called New York, there lived a young man named Joey. Joey was a kind and caring person who always tried to help others. He had a beautiful girlfriend named Rachel, who was also kind and caring. They were in love with each other and wanted to spend their lives together. One day, Joey and Rachel decided to get married. They planned their wedding for a long time, and everyone in the town was excited to attend. On the day of the wedding, Joey and Rachel exchanged vows and promised to love each other forever. After the wedding, Joey and Rachel moved into a beautiful house in the countryside. They were happy and content with their lives, and they had two children, a boy and a girl. The children grew up to be kind and caring people, just like their parents. As the years went by, Joey and Rachel continued to love each other deeply. They had many ups and downs, but they always found a way to overcome their challenges and stay together. Finally, after many years of marriage, Joey and Rachel decided to retire and live in a small cottage in the countryside. They were happy and content with their lives, and they knew that they had found the love of their lives in each other. |

Table 8: Case study on Dream-Instruct-7B. Question: Lily can run 12 kilometers per hour for 4 hours. After that, she runs 6 kilometers per hour. How many kilometers can she run in 8 hours?

| Model | Generate Text (L=128, T=128) |
| --- | --- |
| Dream-Instruct-7B (Entropy) | In the first 4 hours, Lily runs 12 kilometers per hour, so she runs a total of 12 * 4 = 48 kilometers. In the next 4 hours, Lily runs 6 kilometers per hour, so she runs a total of 6 * 4 = 24 kilometers. Therefore, in 8 hours, Lily can run a total of 48 + 24 = 72 kilometers. #### 72 |
| dKV-Cache-Decode, Refresh = 4 | In the first 4 hours, Lily can run 12 kilometers per hour, so she can run a total of 12 * 4 = 48 kilometers. In the next 4 hours, Lily can run 6 kilometers per hour, so she can run a total of 6 * 4 = 24 kilometers. Therefore, in 8 hours, Lily can run a total of 48 + 24 = 72 kilometers. #### 72 |
| dKV-Cache-Decode, Refresh = 16 | In the first 4 hours, Lily runs 12 kilometers per hour, so she runs a total of 12 * 4 = 48 kilometers. In the next 4 hours, Lily runs 6 kilometers per hour, so she runs a total of 6 * 4 = 24 kilometers. Therefore, in 8 hours, Lily can run a total of 48 + 24 = 72 kilometers. The answer is: 72 |
| dKV-Cache-Prefill, No refresh | In the first 4 hours, Lily runs 12 kilometers per hour, so she runs a total of $12 \times 4 = 48$ kilometers. In the next 4 hours, she runs 6 kilometers per hour, so she runs a total of $6 \times 4 = 24$ kilometers. Therefore, in 8 hours, Lily can run a total of $48 + 24 = \backslash boxed\{72\}$ kilometers.The answer is: 72 |

