# OpenReview forum: "dKV-Cache: The Cache for Diffusion Language Models"
_NeurIPS.cc/2025/Conference — NeurIPS 2025 poster_

### Official Review · Reviewer_P5ym · 2025-07-03

**Clarity:** 3
**Significance:** 3
**Originality:** 3
**Rating:** 4
**Confidence:** 4

**Summary:**

This paper addresses the problem of slow inference in Diffusion Language Models (DLMs),  as unlike autoregressive models, DLMs cannot use the standard KV-Cache because their bidirectional attention and non-sequential generation process mean that key and value states are constantly changing for all tokens. The authores propose a new caching method called dKV-Cache to solve this. The core idea is based on their observation that a token's representation becomes stable only after it has been decoded. Therefore, they cache a token's key and value states with a one-step delay, which is critical for maintaining performance. The paper presents two main versions: dKV-Cache-Decode, which offers a large speedup with almost no drop in quality, and dKV-Cache-Greedy, which is even faster but trades a small amount of accuracy for higher speed. Overall, their training-free approach acheives speedup on various benchmarks narrowing the inference speed gap between DLMs and autoregressive models.

**Questions:**

a very interesting result is that dKV-Cache-Decode can improve performance on long sequences (e.g., HumanEval). The paper suggests this might be due to inefficiencies in how bidirectional attention aggregates context. This finding is quite interesting, what is the authors' thoughts on this? Why does freezing older representations help? Does this imply that the model is "overthinking" or that later-stage token refinements are destructively interfering with earlier, stable parts of the sequence?

**Ethical Concerns:**

["NO or VERY MINOR ethics concerns only"]

**Final Justification:**

I remain positive for the final evaluation of this submission, as it's done meaningful and sound contributions for a practical important aspect for dLLMs.

**Limitations:**

yes

**Quality:**

3

**Strengths And Weaknesses:**

Strengths

- The paper is well executed, with a clear motivation and a practical algorithm that fits seamlessly into existing transformer architectures. The experimental protocol is thorough, covering multiple benchmarks (MMLU, GSM8K, HumanEval, MBPP) and two distinct 7B-parameter models (LLaDA, Dream).
- Core ideas—delayed caching, one-step delay, and the two variants (Decode vs. Greedy)—are explained with simple diagrams and step-by-step equations. Readers can follow the transition from standard KV-Cache to the proposed dKV-Cache without undue effort.
- Adapting key–value caching to bidirectional, non-sequential diffusion workflows is novel. The insight that token representations stabilize after decoding—and can thus be safely cached—breaks new ground in diffusion model efficiency.

Weaknesses


- The "2-10x speedup" claim while technically supported by the data, could be nuanced. The 10x (or even 18x) speedup appears in specific long-prefill scenarios, which are important but not universal. In more general cases, the speedup is closer to 2-4x. It would strengthen the paper to more clearly delineate the conditions under which different levels of speedup are achieved, perhaps by discussing the speedup as a function of the prefill-to-generation ratio.

- The chosen baseline (reducing the number of sampling steps) is reasonable as it's a common way to trade quality for speed. However, the paper could be strengthened by briefly discussing how dKV-Cache compares to or complements other DLM acceleration strategies mentioned in the related work, such as semi-autoregressive approaches. This would better position the work within the broader landscape of DLM inference optimization.

- The `concat_reorder` operator is critical for implementing a non-sequential cache efficiently, especially with rotational position embeddings (RoPE). While it's deferred to the appendix, a sentence or two in the main text explaining its function and importance would be beneficial for clarity.

---

> ### Author Rebuttal · Authors · 2025-07-31
>
> We are deeply grateful for the reviewer’s insightful comments and thoughtful suggestions, which have helped us improve the quality of our work. We would revise our manuscript accordingly.
>
> ---
> **W1: Speedup v.s. prefill-to-generation ratio**
> > *W1. The "2-10x speedup" claim while technically supported by the data, could be nuanced. The 10x (or even 18x) speedup appears in specific long-prefill scenarios, which are important but not universal. In more general cases, the speedup is closer to 2-4x. It would strengthen the paper to more clearly delineate the conditions under which different levels of speedup are achieved, perhaps by discussing the speedup as a function of the prefill-to-generation ratio.*
>
>  Thanks for your question and your great suggestion. The acceleration ratio is relevant to both the generation length and also the prefill-to-generation ratio (ratio in the below table). We here give more results about the acceleration ratio regarding this two factors:
>
> * decode length = 32
> | ratio | 0.25 | 0.5 | 1 | 2 | 4 | 8 | 16 |32 | 64 |
> | -- | -- | -- | -- | -- | -- | -- | -- | -- | -- |
> | dKV-Cache-Prefill | 1.15$\times$ | 1.31$\times$ | 1.68$\times$ | 2.34$\times$ | 3.51$\times$ | 5.48$\times$ | 8.21$\times$ | 11.40$\times$ | 13.99$\times$ |
> | dKV-Cache-Decode  | 1.41$\times$ | 1.48$\times$ | 1.66$\times$ | 1.83$\times$ | 2.04$\times$ | 2.18$\times$ | 2.30$\times$ | 2.37$\times$ | 2.43$\times$ |
>
> * decode length = 256
> | ratio |  0.03125 | 0.0625 | 0.125 |0.25 | 0.5 | 1 | 2 | 4 | 8 |
> | -- | -- | -- | -- | -- | -- | -- | -- | -- | -- |
> | dKV-Cache-Prefill  | 1.00$\times$ | 1.03$\times$ | 1.07$\times$ | 1.18$\times$ | 1.39$\times$ | 1.78$\times$ | 2.52$\times$ | 3.92$\times$ | 6.58$\times$ |
> | dKV-Cache-Decode   | 1.39$\times$ | 1.40$\times$ | 1.44$\times$ | 1.52$\times$ | 1.62$\times$ | 1.78$\times$ | 1.97$\times$ | 2.18$\times$ | 2.39$\times$ |
>
> According to the results, dKV-Cache-Prefill can achieve a significantly higher speedup when the prefill-to-generation ratio is large, due to its ability to reuse cached key/value representations for the long prompt. dKV-Cache-Decode is less influenced by the prefill-to-generation ratio.
>
> In our experiments, we extensively evaluated the method under diverse configurations, including a wide range of zero-shot in-context learning settings, to systematically vary the prefill-to-generation ratio and generation length.
>
> ---
>
> **W2: Relationship with semi-autoregressive approaches**
> > *W2. The chosen baseline (reducing the number of sampling steps) is reasonable as it's a common way to trade quality for speed. However, the paper could be strengthened by briefly discussing how dKV-Cache compares to or complements other DLM acceleration strategies mentioned in the related work, such as semi-autoregressive approaches. This would better position the work within the broader landscape of DLM inference optimization.*
>
> Thanks for your valuable suggestion. We identify two key connections between our method and semi-autoregressive approaches:
>
> 1. **Complementarity**. Given a pre-trained semi-autoregressive model (e.g., Block Diffusion[1]), our approach can be seamlessly integrated to provide intra-block KV-Cache, while the semi-autoregressive design itself naturally induces inter-block KV-Cache.
>
> 2. **Transformation from a non-autoregressive DLMs toward semi-autoregressive DLMs**. This is achieved by applying the proposed dKV-Cache-Greedy strategy and replaces random sampling with a deterministic, left-to-right generation order with a fixed-size window. The left-to-right sampling means that we decoded the n-th token at the n-th step, which has the same decoding order as an auto-regressive model, but with a sliding window sliding from left to right in dKV-Cache-greedy. This makes the model act as a semi-autoregressive one, constraining the decoding and the non-cached tokens inside the window.
>
> We conduct exploratory experiments for this (LLaDA-instruct, GSM8K, official performance = 78.6%):
>
> | Decoding Length | Original | dKV-Cache-Greedy (left-to-right sampling) |
> | --| --|  --|
> | 256 | 78.8 (70.0 token/s) | 79.5 (244.4 token/s, 3.5x) |
> | 512 | 81.5 (42.8 token/s) | 83.9 (220.8 token/s, 5.2x) |
> | 1024 | 82.7 (22.4 token/s) | 83.5 (147.4 token/s, 6.6x) |
>
> The experimental results show that with this special sampling strategy under dKV-Cache-Greedy, the model performance remains largely unaffected while enabling the use of a significantly higher cache ratio. This, in turn, leads to a semi-autoregressive model with improved inference speed without compromising accuracy.
>
> We would add this paragraph into the related work section and also add another analysis section for dKV-Cache-Greedy in the revised manuscript.
>
> ---
>
>  **W3: Explanations for concat_reorder**
> > *W3. The concat_reorder operator is critical for implementing a non-sequential cache efficiently, especially with rotational position embeddings (RoPE). While it's deferred to the appendix, a sentence or two in the main text explaining its function and importance would be beneficial for clarity.*
>
> Thanks for your suggestion. We would modify and add this short description about the concat_reorder into Line 184-186 to introduce more about this operator:
>
> The concat_reorder operator first reorders the input sequence by placing cached tokens on the left and newly generated (uncached) tokens on the right before passing it into the model. This reordering affects RoPE, so we also reorder the corresponding rotary position matrix accordingly prior to the model forward pass. This design minimizes token-level reordering overhead by applying it only once per decoding step. During attention computation, each attention module then concatenates the cached and new key/value states, applies the appropriate index reordering, and retrieves the updated cached representations from the combined key/value matrices.
>
> ---
>
>  **Q1: Why dKV-Cache-Decode can improve performance on long sequences**
>
> > *Q1. A very interesting result is that dKV-Cache-Decode can improve performance on long sequences (e.g., HumanEval). The paper suggests this might be due to inefficiencies in how bidirectional attention aggregates context. This finding is quite interesting, what is the authors' thoughts on this? Why does freezing older representations help? Does this imply that the model is "overthinking" or that later-stage token refinements are destructively interfering with earlier, stable parts of the sequence?*
>
> Thanks for your insightful question. We also think that this phenomenon is very interesting.
>
> This observation, as you noted, suggests that updating a token and thereby updating the key and value states can in fact degrade the prediction quality of future tokens. We attribute this to **error accumulation during intermediate generation stages**, where incorrectly generated tokens propagate and negatively impact subsequent decoding. Unlike some autoregressive models with potential for correction (e.g., reflection), current DLMs ( LLaDA and Dream ) **lack any refinement mechanism**, meaning that early mistakes cannot be revised and persist throughout the sequence. The cache mechanism helps mitigate this issue by using the previous key and value states, effectively bypassing the influence of erroneous tokens and leading to more stable decoding.
>
>
> [1] Block Diffusion: Interpolating Between Autoregressive and Diffusion Language Models

---

> > ### Comment · Reviewer_P5ym · 2025-08-06
> >
> > Thanks authors for the detailed response which helped addressing my questions. I'll keep my positive evaluation for this work.

---

> > > ### Author Response · Authors · 2025-08-06
> > >
> > > We sincerely appreciate your insightful suggestions and comments. We will polish our paper according to your suggestions.

---

### Official Review · Reviewer_2PJU · 2025-07-03

**Clarity:** 4
**Significance:** 2
**Originality:** 2
**Rating:** 3
**Confidence:** 4

**Summary:**

The paper introduces a caching mechanism, dKV-Cache, tailored for Diffusion Language Models (DLMs), which traditionally lack the ability to reuse key/value states during inference. The method is motivated by the empirical observation that token representations tend to stabilize after decoding, allowing delayed reuse of key and value vectors.

Two variants are proposed:

dKV-Cache-Decode: A one-step delayed caching strategy that prioritizes accuracy and stability.

dKV-Cache-Greedy: A more aggressive approach using a local window around the current token to reduce complexity from O(L³) to O(L²), trading off some accuracy for greater speed.


Extensive experiments on benchmarks such as MMLU, GSM8K, and HumanEval show 2–10× inference speedups with minimal performance degradation. The method requires no model retraining, making it a lightweight and practical optimization for accelerating inference in existing DLMs.

**Questions:**

see cons

**Ethical Concerns:**

["NO or VERY MINOR ethics concerns only"]

**Final Justification:**

Thank you for the authors’ detailed and thoughtful responses, which addressed several of my earlier points. However, my primary concern remains, and I am still not fully convinced about the level of novelty. Accordingly, I will keep my score unchanged.

**Limitations:**

yes

**Quality:**

2

**Strengths And Weaknesses:**

Pros

1. Well-defined research problem: The paper addresses a fundamental bottleneck in diffusion language models (DLMs)—the lack of key-value caching—which is the main reason for their slow inference compared to autoregressive models.


2. Simple yet practical idea: The proposed method is straightforward and easy to implement. It provides a lightweight inference-time optimization without requiring any retraining or architectural changes.


3. Clear writing and presentation: The paper is well-organized and the method is clearly explained, making it easy to follow and reproduce.


4. Broad applicability: The technique works across various tasks (e.g., QA, math, code), different model sizes, and generation lengths, showing strong robustness and generality.




---

Cons

1. Lack of theoretical justification: The method is based purely on empirical observations rather than formal theoretical guarantees. In particular, the reuse of key/value states lacks a rigorous foundation, unlike AR models where caching is mathematically supported by causal masking. The use of the term “KV-Cache” may therefore be misleading.


2. Method is somewhat incremental: Similar ideas (e.g., exploiting representation stability across steps) have already appeared in the diffusion literature, particularly in text-to-image generation. The contribution here mainly lies in applying them to DLMs.


3. Missing comparison to AR models: The paper does not show how the proposed caching mechanism compares against autoregressive models in terms of both throughput and generation quality, which is crucial for justifying practical relevance.


4. No architectural improvements for lossless caching: The work focuses solely on inference-time heuristics. A more ambitious direction—modifying DLM architecture to enable theoretically lossless caching—would have been more compelling and impactful.

---

> ### Author Rebuttal · Authors · 2025-07-31
>
> We are thankful for the reviewer’s insightful comments and suggestions, and we will incorporate the following analysis and provide further clarification in the revised manuscript.
>
> ------
> **W1: Lack of theoretical justification**
> > *W1: The method is based purely on empirical observations rather than formal theoretical guarantees. In particular, the reuse of key/value states lacks a rigorous foundation, unlike AR models where caching is mathematically supported by causal masking. The use of the term “KV-Cache” may therefore be misleading.*
>
>
> Thank you for the insightful question. While diffusion language models do not naturally admit autoregressive causal masking, we provide a theoretical justification for the validity of reusing stale key/value states from earlier steps
>
> Our intuition is to treat the cached key/value states as propagation states across decoding steps, similar to the hidden states in RNNs or in linearized attention variants. We aim to derive the bound if we reuse the key/value states from previous steps, e.g., using $k_i^{t-k}$ and $v_i^{t-k}$ at step $t$ instead of freshly computed ones $k_i^{t}$ and $v_i^{t}$. $L$ below means the sequence length.
>
> Consider the standard attention output at decoding step $t$. Following prior work on kernelized attention, we approximate the softmax with a positive kernel $\phi(\cdot)$:
>
> $$
> o_t=\frac{\phi\left(q_t\right) \sum_{i=1}^L \phi\left(k_i^t\right)^{\top} v_i^t}{\phi\left(q_t\right) \sum_{i=1}^L \phi\left(k_i^t\right)^{\top}}
> $$
> And we have:
>
> $$
> Z_t = \sum_{i=1}^L \phi(k_i^t), P_t = \sum_{i=1}^L \phi(k_i)^\top v_i^t, o_t = \frac{\phi(q_t)P_t}{\phi(q_t)Z_t}
> $$
>
> Here for dKV-Cache:
> * Some $k_i^t$ and $v_i^t$ are fresh, while others are replaced with cached $\tilde{k}_i^t = {k}_i^{t-k}$ and $\tilde{v}_i^t = {v}_i^{t-k}$.
> * We denote $\tilde{P}_t$ as $P_t$ cached counterpart; similarly $\tilde{Z}_t$ and $Z_t$.
>
> Let’s define the output error for attention as: $\delta_t=\left\|| o_t- \tilde{o}_t\right\|| $, we would get:
>
> $$
> \delta_t = \left\|| o_t- \tilde{o}_t\right\|| =  \left\|\left\|  \frac{\phi(q_t)P_t}{\phi(q_t)Z_t} - \frac{\phi(q_t)\tilde{P}_t}{\phi(q_t)\tilde{Z}_t} \right \|\right \|
> \leq \frac{\left\||\phi\left(q_t\right)\right\|| \cdot || P_t-\tilde{P}_t || }{\left|\phi\left(q_t\right) Z_t\right|} + \left\||\phi\left(q_t\right)\right\|| \cdot || \tilde{P}_t || \cdot\left|\frac{1}{\phi\left(q_t\right) Z_t}-\frac{1}{\phi\left(q_t\right) \tilde{Z}_t}\right|
> $$
>
> First, for
>
> $$
> || P_t-\tilde{P_t} || =\left\|\left|\sum_{i=1}^L\left(\phi\left(k_i^t\right)^{\top} v_i^t-\phi\left( \tilde{k}_i^t\right)^{\top} \tilde{v}{ }_i^t\right)\right\|\right|
> $$
>
> Let’s denote $\mathcal{M}_t \subseteq \\{1,\dots,t\\}$ as the set of indices for which cached states are used, i.e., $\tilde{k}_i^t = k_i^{t-k}$ and $\tilde{v}_i^t = v_i^{t-k}$. For fresh tokens ($i \notin \mathcal{M}_t$), $\tilde{k}_i = k_i^t$ and $\tilde{v}_i = v_i^t$, so the difference is zero. The sum becomes:
>
> $$
> || P_t-\tilde{P_t} || =\left\|\left\| \sum_{i \in \mathcal{M}_t}^L\left(\phi\left(k_i^t\right)^{\top} v_i^t-\phi\left( \tilde{k}_i^t\right)^{\top} \tilde{v}{ }_i^t\right)\right\|\right\|   \leq \left\| \mathcal{M}_t \right\| \cdot (C_k\epsilon_v + C_v\epsilon_k)
> $$
>
> with $||\phi(k_i^t) - \phi(\tilde{k}_i)|| \leq \epsilon_k$, $||v_i^t - \tilde{v}_i|| \leq \epsilon_v$, $||\phi(k_i^t)|| \leq C_k$, $||\tilde{v}_i|| \leq C_v$.
>
> For the denominator drift part, we aim to bound:
>
> $$
> \left|\frac{1}{\phi\left(q_t\right) Z_t}-\frac{1}{\phi\left(q_t\right) \tilde{Z}_t}\right|=\left|\frac{\phi\left(q_t\right)\left(\tilde{Z}_t-Z_t\right)}{\phi\left(q_t\right) Z_t \cdot \phi\left(q_t\right) \tilde{Z}_t}\right|
> $$
>
> Assume $\phi(q_t) Z_t \geq \gamma$ and $\phi(q_t) \tilde{Z}_t \geq \gamma'$, similar to the previous one, we can get
>
> $$
> \left|\frac{1}{\phi\left(q_t\right) Z_t}-\frac{1}{\phi\left(q_t\right) \tilde{Z}_t}\right| \leq \frac{\left\||\phi\left(q_t\right)\right\|| \cdot\left\||\Delta Z_t\right\||}{\gamma \cdot \gamma^{\prime}} \leq \frac{\left\||\phi\left(q_t\right)\right\|| \cdot\left\|\mathcal{M}_t\right\| \cdot \epsilon_k}{\gamma \cdot \gamma^{\prime}}
> $$
>
> Thus, summarizing the above two, we would get $\delta_t$ is bounded by
>
> $$
> \delta_t \leq \frac{||\phi(q_t)|| \cdot\left|\mathcal{M}_t\right|\left(C_v \epsilon_k+C_k \epsilon_v\right)}{\gamma} + \frac{\left\||\phi\left(q_t\right)\right\||^2 \cdot ||\tilde{P}_t|| \cdot\left\|\mathcal{M}_t\right\| \cdot \epsilon_k}{\gamma \cdot \gamma^{\prime}}
> $$
>
> **Interpretation of the bound**:
> * The more stable the cached $k, v$, the smaller $\epsilon_k, \epsilon_v$ → smaller error.
> * The fewer tokens we cache ($|\mathcal{M}_t|$), the smaller the error.
> * If dKV-Cache avoids caching tokens (how to select $\mathcal{M}_t$) with high dynamics (i.e., large $|\phi(k_i^t) - \phi(k_i^{t-k})|$), the smaller the error.
>
> ---
> **W2: Method is somewhat incremental**
> > *W2: Similar ideas (e.g., exploiting representation stability across steps) have already appeared in the diffusion literature, particularly in text-to-image generation. The contribution here mainly lies in applying them to DLMs.*
>
> Thank you for your question. While our method does use the representation stability across steps, the core contribution lies in the **delayed cache mechanism**, which is specifically designed for discrete diffusion language models. **This delayed caching strategy is unique to the discrete nature of text-based diffusion and is not applicable to standard text-to-image diffusion models**. This comes from the special evolving dynamics for the DLMs, and importantly, this design has a substantial impact on performance. As illustrated in Figure 3, when the cache ratio is fixed at 48\%, **the model’s accuracy improves dramatically, from 3.33\% to 74.3\%**, when using the delayed cache compared to a naïve caching strategy.
>
> Building on this, we propose two variants for dKV-Cache: the greedy one, which uses a tolerant window to allow approximate caching, and the decode one, which aligns caching with decoding dynamics for better performance. **Both of these two designs are designed for discrete diffusion language models and cannot be used on image diffusion models.**
>
> ---
> **W3: Missing comparison to AR models**
> > *W3: The paper does not show how the proposed caching mechanism compares against autoregressive models in terms of both throughput and generation quality, which is crucial for justifying practical relevance*
>
> Thanks for your great question. Despite incorporating dKV-Cache, current Diffusion Language Models still fail to outperform AR models in terms of decoding speed. Below, we present a speed and complexity analysis comparing LLaDA and LLaMA3 under two scenarios: long decoding length (L=1024) and short decoding length (L=32).
>
> * Long Decoding (L=1024, zero-shot, LLaDA-Instruct)
>
> | | Token/s | Accuracy |
> | -- | -- | -- |
> | LLaDA (entropy-based sampling)  |  22.4 | 82.1 |
> | LLaDA w/ dKV-Cache-decode  |  56.0 | 82.5 |
> | LLaDA (random sampling)  |  23.2 | 79.5 |
> | LLaDA w/ dKV-Cache-Greedy  |  91.7 | 77.4 |
> | LLaMA3-8B  | 1463.9  | 73.7 |
>
> * Short Decoding (L=32, zero-shot, LLaDA-Instruct)
>
> |  | Token/s | Accuracy |
> | -- | -- | -- |
> | LLaDA  |  149.7 |  38.6  |
> | LLaDA with dKV-Cache-decode |  230.4 | 38.4 |
> | LLaDA (random sampling) | 151.1 |  34.4  |
> | LLaDA with dKV-Cache-greedy  | 190.1 | 28.2 |
> | LLaMA-8B  | 1524.9 | 1.74 |
>
> * FLOPs analysis
>
> We provide a FLOPs analysis for attention computation per token during generation:
> 1. For DLMs, the FLOPs is $ 4L^2d + 4Ld^2$ per token
> 2. For ARs, the FLOPs is $ 4Ld + 4d^2$ per token
> 3. With dKV-Cache-decode, the expected FLOPs per token is: $ 2L^2d + 2Ld^2$
> 4. With dKV-Cache-greedy, the FLOPs is $ 4wLd + 4wd^2$, where $w$ denotes the attention window size (typically $w \leq 4$).
>
> This indicates that **unaccelerated DLMs require approximately $L\times$ more FLOPs than ARs for attention computation**, leading to significantly slower inference on long sequences. The proposed **dKV-Cache-decode** method mitigates this overhead by reducing the FLOPs **by half**, while **dKV-Cache-greedy** further improves efficiency by reducing the cost **by a factor of L/w**.
>
>
> **Summarization**:
>
> 1. Despite utilizing dKV-Cache to narrow the gap, **LLaDA still lags significantly behind ARs in terms of decoding speed**, although it achieves competitive performance in terms of generation quality.
>
> 2. One scenario where **DLMs outperform AR models is under short decoding lengths**. Specifically, when the decoding length is limited to L=32, AR models like LLaMA struggle to produce coherent or complete answers, achieving only a score of 1.74, whereas LLaDA maintains strong performance with a score of 38.7. This demonstrates that DLMs are more effective under constrained generation budgets.
>
> **Solutions**:
>
> To make DLMs as fast or faster than ARs, several key components are needed:
>
> 1. dKV-Cache-Greedy to match ARs' complexity with respect to sequence length $L$.
> 2. Parallel decoding to reduce the number of sampling steps.
> 3. Semi-autoregressive decoding to bound $L$ and limit computation.
> 4. System-level optimizations such as vLLM and FlashAttention for improved runtime efficiency.
>
> ---
> **W4: No architectural improvements for lossless caching**
> > *The work focuses solely on inference-time heuristics. A more ambitious direction—modifying DLM architecture to enable theoretically lossless caching—would have been more compelling and impactful*
>
>
> Thanks for your suggestion. We fully agree that this problem maybe can be resolved from the model architecture, such as by redesigning the attention mechanism or the causal mask in a way that ensures strict equivalence for attention calculation and then pretrain this model. It would be highly interesting, impactful and also an ambitious idea and of significant theoretical and practical value. We hope to explore it in the future, making the architecture of DLMs to natively support the KV-Cache mechanisms.

---

> ### Author Response · Authors · 2025-08-07
> **We Look Forward to Hearing Your Thoughts**
>
> Dear Reviewer 2PJU,
>
> Thank you again for your thoughtful and constructive comments, and also for the time and effort in reviewing our paper.
>
> We sincerely look forward to hearing any further insightful comments you may have. We would greatly appreciate it if you could let us know whether any concerns may need further clarification. We are more than happy to provide additional explanations.
>
> Best Regards,
> Authors of Submission 3030

---

### Official Review · Reviewer_vSQt · 2025-07-03

**Clarity:** 2
**Significance:** 3
**Originality:** 3
**Rating:** 5
**Confidence:** 3

**Summary:**

KV-Cache is a widely adopted technique for accelerating sampling in large language models (LLMs). However, the recently emerging Diffusion Language Models (DLMs) do not support KV-Cache. This paper analyzes the reasons behind this limitation and systematically investigates the dynamic behaviors that occur during the sampling process of DLMs. Based on these observations, the authors propose the dKV-Cache mechanism. Experiments on two popular open-source models, LLaDA and Dream, as well as a wide range of downstream tasks, demonstrate that dKV-Cache can boost inference speed by 2–10×.

**Questions:**

1. Could you provide more explanation for Figure 2(b)? It appears that the feature distances both before and after decoding are relatively high and fairly similar. However, the paper claims that representations become stable after decoding, while they fluctuate more before decoding. I find this somewhat confusing and would appreciate a more detailed clarification.

2. Why do LLaDA and Dream use different evaluation settings? LLaDA is evaluated on an A6000 GPU, while Dream is evaluated on an H20. Also, since Dream uses a few-shot in-context learning setup, did LLaDA use few-shot as well? Moreover, why does this paper use the Instruct model for LLaDA but the base model for Dream?

3. Could you clarify what T refers to in Table 2? It doesn't seem to represent the number of sampling steps. Also, why is T not reported in Table 1?

**Ethical Concerns:**

["NO or VERY MINOR ethics concerns only"]

**Final Justification:**

I consider this work a new attempt at adding cache to diffusion language models. Most of my concerns have been addressed in the rebuttal. However, I remain doubtful about some experimental results, which appear inconsistent with concurrent studies (see Replying to Rebuttal by Authors). Therefore, I will keep my score at 4.

**Limitations:**

yes

**Quality:**

3

**Strengths And Weaknesses:**

## Strengths
1. This paper is the first to demonstrate that KV-Cache can be integrated into DLMs in a training-free manner, which is a novel and meaningful contribution.

2. The paper begins with a careful analysis of the unique characteristics of the DLM sampling process and then designs the method accordingly—the motivation is clear and well-justified.

3. The proposed method shows consistent improvements across both open-source models (LLaDA and Dream) and a wide range of tasks, including mathematical reasoning and code generation.

## Weaknesses
1. The paper lacks analysis of some key factors, such as the effect of batch size on acceleration. I’m particularly curious how the speedup of dKV-Cache changes under varying batch sizes (i.e., different GPU utilization levels).

2. The paper does not include comparisons with autoregressive baselines. After applying dKV-Cache, is the sampling speed of DLMs actually faster than that of autoregressive models?

---

> ### Author Rebuttal · Authors · 2025-07-31
>
> We deeply appreciate the reviewer’s constructive feedback, and we will revise the manuscript accordingly
>
> ---
> **W1: Lack of analysis of batch size**
>
> > *W1: The paper lacks analysis of some key factors, such as the effect of batch size on acceleration. I’m particularly curious how the speedup of dKV-Cache changes under varying batch sizes (i.e., different GPU utilization levels).*
>
> Thanks for your great question. We also think this is a crucial problem and thus we have conducted an analysis on the effects of batch size and decoding length in Appendix D. Please refer to Figure 7 in the Appendix for full details. Below, we summarize the key observations and conclusions:
>
> * At low GPU utilization (e.g., small batch size or short decoding length), the GPU is underutilized, and therefore the speedup is limited.
>
> * When batch size = 1, the improvement in latency is marginal due to insufficient utilization of the GPU.
>
> * When batch size >= 2, the speedup becomes pronounced, reaching approximately 1.4×.
>
> * When batch size >= 64, the acceleration ratio stabilizes at around 2.5×, indicating that our method has better speedup with full GPU utilization.
>
> | Method | bsz=1 | bsz = 2 | bsz = 4 | bsz = 16 | bsz = 64 | bsz = 256 |
> | --     | -- |  -- |  -- |  -- | -- |  -- |
> | Origin           | 23.1 | 30.7 | 44.9 | 56.6  | 59.7  | 60.7 |
> | dKV-Cache-Decode | 24.5 (1.06x) | 43.3 (1.41x) | 71.9 (1.60x) | 121.3 (2.14x) | 145.7 (2.44x) | 153.3 (2.52x) |
>
> ---
> **W2:  Comparisons with autoregressive baselines**
>
> > *W2. The paper does not include comparisons with autoregressive baselines. After applying dKV-Cache, is the sampling speed of DLMs actually faster than that of autoregressive models?*
>
> Thanks for your great question. Despite incorporating dKV-Cache, current Diffusion Language Models still fail to outperform AR models in terms of decoding speed. Below, we present a speed and complexity analysis comparing LLaDA and LLaMA3 under two scenarios: long decoding length (L=1024) and short decoding length (L=32).
>
> * Long Decoding (L=1024, zero-shot, LLaDA-Instruct)
>
> |   | Token/s | Accuracy |
> | -- |  -- |  -- |
> | LLaDA (entropy-based sampling)  |  22.4      | 82.1 |
> | LLaDA w/ dKV-Cache-decode       |  56.0      | 82.5 |
> | LLaDA (random sampling)         |  23.2      | 79.5 |
> | LLaDA w/ dKV-Cache-Greedy       |  91.7      | 77.4 |
> | LLaMA3-8B                       |  1463.9    | 73.7 |
>
> * Short Decoding (L=32, zero-shot, LLaDA-Instruct)
>
> |   | Token/s | Accuracy |
> | -- |  -- |  -- |
> | LLaDA                        |  149.7 |  38.6  |
> | LLaDA with dKV-Cache-decode  |  230.4 |  38.4  |
> | LLaDA   (random sampling)    |  151.1 |  34.4  |
> | LLaDA with dKV-Cache-greedy  |  190.1 |  28.2  |
> | LLaMA-8B                     |  1524.9 |  1.74 |
>
> * FLOPs analysis
>
> We provide a FLOPs analysis for attention computation per token during generation:
> 1. For DLMs, the FLOPs is $ 4L^2d + 4Ld^2$ per token
> 2. For ARs, the FLOPs is $ 4Ld + 4d^2$ per token
> 3. With dKV-Cache-decode, the expected FLOPs per token is: $ 2L^2d + 2Ld^2$
> 4. With dKV-Cache-greedy, the FLOPs is $ 4wLd + 4wd^2$, where $w$ denotes the attention window size (typically $w \leq 4$).
>
> This indicates that **unaccelerated DLMs require approximately $L\times$ more FLOPs than ARs for attention computation**, leading to significantly slower inference on long sequences. The proposed **dKV-Cache-decode** method mitigates this overhead by reducing the FLOPs **by half**, while **dKV-Cache-greedy** further improves efficiency by reducing the cost **by a factor of L/w**.
>
>
> **Summarization**:
>
> 1. Despite utilizing dKV-Cache to narrow the gap, **LLaDA still lags significantly behind ARs in terms of decoding speed**, although it achieves competitive performance in terms of generation quality.
>
> 2. One scenario where **DLMs outperform AR models is under short decoding lengths**. Specifically, when the decoding length is limited to L=32, AR models like LLaMA struggle to produce coherent or complete answers, achieving only a score of 1.74, whereas LLaDA maintains strong performance with a score of 38.7. This demonstrates that DLMs are more effective under constrained generation budgets.
>
> **Solutions**:
>
> To make DLMs as fast or faster than ARs, several key components are needed:
>
> 1. dKV-Cache-Greedy to match ARs' complexity with respect to sequence length $L$.
> 2. Parallel decoding to reduce the number of sampling steps.
> 3. Semi-autoregressive decoding to bound $L$ and limit computation.
> 4. System-level optimizations such as vLLM and FlashAttention for improved runtime efficiency.
>
> ---
> **Q1: Explanation for Figure 2(b)**
> > *Q1：Could you provide more explanation for Figure 2(b)? It appears that the feature distances both before and after decoding are relatively high and fairly similar. However, the paper claims that representations become stable after decoding, while they fluctuate more before decoding. I find this somewhat confusing and would appreciate a more detailed clarification.*
>
> Sorry for the unclear explanation here. For Figure 2(b), the calculation pipeline is as follows:
>
> 1. We compute the pairwise K/V distance between two adjacent decoding steps, resulting in an array dis[:] of length T-1 for a sequence of T steps.
>
> 2. To analyze the impact of decoding for the i-th token, we compute:
>       * Before Distance: avg(dis[:i]),
>       * After Distance: avg(dis[i:])
>
> From this, we obtain the curve shown in Figure 2(b). The key observations are:
>
> * The **before-decoding distance is generally larger than the after-decoding distance**, suggesting that the hidden states become more stable as decoding progresses.
>
> * We observe a fluctuation pattern, and in particular, the last step shows after-distance > before-distance. This is because the maximum change in hidden states occurs exactly at the step when a token is decoded. Hence, the ranking of distance magnitudes is:
>               at decoded step > before decode > after decode
>
> For the last token, there’s no step beyond it, so avg(dis[-1:]) = dis[-1], which is exactly at the decode step and has a large value.
> We think that there is room to improve the presentation of Figure 2(b). Following your suggestion, we plan to enhance it by adding separate curves for before, at decoded, and after distances. Thank you for the valuable feedback.
>
> ---
> **Q2:  LLaDA and Dream with different evaluation settings**
> > *Q2. Why do LLaDA and Dream use different evaluation settings? LLaDA is evaluated on an A6000 GPU, while Dream is evaluated on an H20. Also, since Dream uses a few-shot in-context learning setup, did LLaDA use few-shot as well? Moreover, why does this paper use the Instruct model for LLaDA but the base model for Dream?*
>
> Thanks for your question. Since there is no big difference between using a base/instruct model regarding the proposed dKV-cache algorithm, we choose the following two settings for Dream and LLaDA in our submission to cover as many cases as possible (whether ICL, Instruct/base, different prefill length):
> * LLaDA: LLaDA-Instruct, no in-context learning, A6000.
> * Dream: Dream-base, in-context learning, H20.
>
> Here we also show another group of results on Dream-Instruct-7B, with similar experimental results as in Dream-base-7B (no ICL, H20, same hyper-parameters as in Dream-Base-7B).
>
> |         |    Origin  |   Half-Steps  | dKV-Cache-Decode   |   dKV-Cache-Prefill        |
> | -- |  -- |  -- |  -- |  -- |
> | GSM8K     |     81.8    |    75.9    |         81.2         |         79.9 |
> | MBPP     |      58.2  |      43.8    |         57.0       |           56.2 |
> | HumanEval  |    52.7  |      28.7    |         53.0    |              57.3 |
> | MMLU     |      69.7   |     69.7     |        69.7       |           69.8 |
> | GPQA    |       31.5   |     30.4     |        30.8     |             30.8 |
>
> We provide here another comparison of Dream and LLaDA on the same device (H20) to directly compare their throughput (token/s), where LLaDA-8B is slightly slower than Dream-7B.
>
> | Decode Length | 32 | 64 | 128 | 256 |
> | -- | -- | -- | -- | -- |
> | LLaDA-8B | 82.9 | 60.8 | 41.4 | 24.6 |
> | Dream-7B | 86.9 | 63.8 | 42.1 | 25.1 |
>
> ---
> **Q3: What T refers to in Table 2**
> >*Q3. Could you clarify what T refers to in Table 2? It doesn't seem to represent the number of sampling steps. Also, why is T not reported in Table 1?*
>
> Thank you for your question and for pointing out the unclear expressions. In Table 2, T refers to the total number of decoding steps (as defined in Line 123). We will clarify this in the caption of the table for better readability.
>
> Regarding Table 1, the experimental setup follows the hyperparameter configuration used in LLaDA, which involves configuring T(sampling steps), L(decoding length), block size, cache interval and window size. For clarity, we listed the detailed hyperparameters, including T, in Table 5 in the appendix.

---

> > ### Comment · Reviewer_vSQt · 2025-08-04
> >
> > Thank you for your response. I believe that most of my concerns have been directly addressed. However, I remain skeptical about the claim that LLaMA3 achieves over 10× higher throughput than Diffusion-LLM (which already incorporates caching). This result appears to differ significantly from the conclusions reported in contemporaneous works [1, 2]. Taking into account both the rebuttal and the other reviewers' comments, I have decided to keep my score.
> >
> > [1] Wu et al. Fast-dLLM: Training-free Acceleration of Diffusion LLM by Enabling KV Cache and Parallel Decoding.
> >
> > [2] Wei et al. Accelerating Diffusion Large Language Models with SlowFast Sampling: The Three Golden Principles.

---

> > > ### Author Response · Authors · 2025-08-06
> > >
> > > We sincerely thank the reviewer for the thoughtful comments and follow-up.
> > >
> > > ---
> > >
> > > * **This result appears to differ significantly from the conclusions reported in contemporaneous works**
> > >
> > > The discrepancy in acceleration conclusions between our work and contemporaneous studies primarily arises from **the choice of batch size during evaluation**. Specifically, those studies used a batch size of 1, under which **ARs cannot fully utilize GPU resources but DLMs reach much higher GPU utilizations**, resulting in low throughput for ARs (e.g., 37 tokens/sec reported in [4], 33.79 tokens/sec in [5], compared to 1524 tokens/sec in our submission).
> > >
> > > We would like to provide evidence from official technical blogs and model reports to illustrate how inference throughput varies with batch size for the LLaMA-8B model:
> > >
> > > | Method | Model | Throughput |
> > > |--|--|--|
> > > | Fast-dLLM, bsz = 1 | LLaMA-3-8B | 37 tokens / sec on A100s |
> > > | Ours, bsz=1 | LLaMA-3-8B | 131 tokens / sec on H100s |
> > > | Ours, bsz=32 | LLaMA-3-8B | 1524 tokens/ sec on H100s|
> > > |||
> > > | Pytorch Official[1], bsz = 1  |  LLaMA 3.1-8B | 131 tokens/sec on H100s|
> > > | Pytorch Official[1], bsz = 32 |  LLaMA 3.1-8B | 5575 tokens/sec on H100s |
> > > | TensorRT-LLM Official[2], bsz = 1 | GPT-J-6B | 185 tokens/sec on H100s|
> > > | TensorRT-LLM Official[2], bsz = 64 | GPT-J-6B | 10907 tokens/sec on H100s |
> > > | LLaMA technical report[3], bsz = 1 | LLaMA-405B | 143 tokens / sec on 16 H100s |
> > > | LLaMA technical report[3], bsz = 1 | LLaMA-405B | 1090 tokens / sec on 16 H100s |
> > >
> > > Papers [4,5] report results using a batch size of 1, under which ARs cannot fully utilize GPU resources, whereas DLMs reach much higher GPU utilizations. **This is evidenced by the result that increasing the batch size significantly improves AR throughput (up to 11.6×), while the throughput gain for LLaDA is much smaller (2.6×)**. Our reported results adopt a large batch size to ensure full GPU utilization for both AR and DLM models, which enables ARs to have a much faster speed but not for DLMs.
> > >
> > > We think that KV-Cache is one of the key optimizations and components enabling DLMs to match or exceed AR performance. More acceleration techniques need to be integrated in DLMs, such as reducing the number of sampling steps, enabling parallel decoding, optimizing inference engines (e.g., FlashAttention, paged attention), and advanced model serving techniques.
> > >
> > > [1] Pytorch Official Blog. Blog Title: Accelerating LLM Inference with GemLite, TorchAO and SGLang.
> > > [2] TensorRT-LLM Official Blog. Blog Title: H100 has 4.6x A100 Performance in TensorRT-LLM, achieving 10,000 tok/s at 100ms to first token.
> > > [3] The Llama 3 Herd of Models. Figure 24 (right).
> > > [4] Fast-dLLM: Training-free Acceleration of Diffusion LLM by Enabling KV Cache and Parallel Decoding.
> > > [5] Accelerating Diffusion Large Language Models with SlowFast Sampling: The Three Golden Principles.

---

> > > > ### Comment · Reviewer_vSQt · 2025-08-07
> > > >
> > > > Thank you for the response. It helped further clarify my remaining questions. I hope the paper will be accepted. Considering the scores from the other reviewers, I have decided to increase my score to 5.

---

> > > > > ### Author Response · Authors · 2025-08-07
> > > > >
> > > > > We sincerely thank the reviewer for the thoughtful feedback and the kind decision to raise the score. We will revise the manuscript accordingly, including a more comprehensive comparison with ARs, an in-depth analysis of acceleration ratios, and improved writing with clearer explanations of the figures and evaluation settings.

---

### Official Review · Reviewer_TZvD · 2025-07-03

**Clarity:** 4
**Significance:** 3
**Originality:** 4
**Rating:** 4
**Confidence:** 4

**Summary:**

This paper proposes a KV-Cache mechanism specifically designed for Diffusion Language Models (DLMs) to address their slow inference problem. The authors introduce "delayed KV-Cache" (dKV-Cache), which enables caching of key-value states in bidirectional attention models by leveraging the observation that token representations stabilize after being decoded. The method achieves 2-10× speedup with minimal performance degradation across various benchmarks.

**Questions:**

See weaknesses

**Ethical Concerns:**

["NO or VERY MINOR ethics concerns only"]

**Final Justification:**

Based on the information, I think it is a valuable work for accelerating dLLMs. Therefore, I think a weak accept should be good.

**Limitations:**

See weaknesses

**Quality:**

3

**Strengths And Weaknesses:**

**Strengths:**
1. This paper focuses on a practical problem of DLMs, which show promise but suffer from significantly slower inference compared to autoregressive models due to the lack of KV cache. Therefore, DLMs require significantly higher computational costs.
2. The paper provides a clear identification of why standard KV-Cache fails for DLMs and an empirical analysis of token representation dynamics throughout the diffusion process.
3. The delayed caching strategy is intuitive and well-motivated while two complementary variants (dKV-Cache-Decode and dKV-Cache-Greedy) provide different trade-offs. Moreover, all methods require no further training
4. Experiments show the practical acceleration over existing SOTA DLMs on various tasks.

**Weaknesses:**
1. The experiments remain somewhat confusing, as they utilize different variants and baselines across various figures for different base models. I would expect the authors to explain why we can only combine dKV-Cache-Greedy with the random version of LLada and why it cannot be used for Dream.
2. From the experiment results, it appears that dKV-Cache-Greedy still works in limited cases; therefore, it remains a question of how to reduce complexity while maintaining performance in real-world applications.

---

> ### Author Rebuttal · Authors · 2025-07-30
>
> We sincerely thank the reviewer for the constructive and valuable feedback. We would revise and include the below experiments and analysis in our manuscript.
>
> ---
>
> **W1: Why cannot dKV-Cache-Greedy be used on Dream?**
>
> > *W1: The experiments remain somewhat confusing, as they utilize different variants and baselines across various figures for different base models. I would expect the authors to explain why we can only combine dKV-Cache-Greedy with the random version of LLada and why it cannot be used for Dream.*
>
> Thanks for your question. dKV-Cache-Greedy can also be applied to Dream, and we provide the corresponding results below. We selected hyperparameters for dKV-Cache-Greedy to achieve approximately 2× acceleration.
>
> | |            Entropy |   Random  |  Random w/ dKV-Cache-Greedy  |
> | --|  --|  --|  --|
> | gsm8k    |    76.88  |   8.11    |   8.58    |
> | humaneval |   57.93  |   1.22   |    1.83  |
> | mbpp      |   55.8   |  17.4   |    17.4        |
> | math       |  39.92  |  4.02    |   4.20         |
> | GPQA     |    36.83 |   33.7  |     34.15      |
>
> Dream's **random sampling yields significantly worse performance compared to entropy-based sampling**. In some datasets, we observed over a 60\% drop in accuracy, with overall accuracy falling below 10\%. We hypothesize that this is because DREAM is adapted from an auto-regressive model and thus inherently favors left-to-right generation. As a result, it is less compatible with random sampling strategies.
>
> We think evaluating our method on top of DREAM with random sampling makes it difficult to draw some reliable and generalized conclusions to show the effectiveness of our approach. Thus, we didn't put the paper in the main body of the paper. We would add these results in the Appendix.
>
> ---
>
> **W2: dKV-Cache-Greedy still works in limited cases**
> > *W2: From the experiment results, it appears that dKV-Cache-Greedy still works in limited cases; therefore, it remains a question of how to reduce complexity while maintaining performance in real-world applications.*
>
> Thank you for your insightful question. We agree that dKV-Cache-Greedy, as presented in this paper, has limitations in real-world applications. We first identify that the core reason for these limited cases comes from **the unknown decoding order** and then propose some potential ways to solve this problem.
>
> * **Problem: Unknown decoding order limits the application of dKV-Cache-Greedy**
>
> Here, we try another sampling method, AR sampling, to decode LLaDA-instruct. AR sampling refers to enforcing a deterministic left-to-right decoding order, just as the AR models. Here are the experimental results on GSM8K:
>
> | Method | Token/s | Accuracy |
> | --- | --- | --- |
> |LLaDA (AR sampling)             |  24.6              | 82.8 |
> |LLaDA w/ dKV-Cache-Greedy       |  174.4 (7.08x)     | 83.5 (+0.7) |
> ||||
> |LLaDA (random sampling)         |  23.2              | 79.5 |
> |LLaDA w/ dKV-Cache-Greedy       |  91.7 (3.95x)      | 77.4 (-2.1)|
>
> Under this sampling method, we observe that model performance remains largely unaffected while enabling the use of a significantly higher cache ratio (less refresh of the cache). This, in turn, leads to improved inference speed without compromising accuracy.
>
> * **Solution: How to know the decoding order in advance?**
>
> Thus, the primary challenge lies in its reliance on knowing the decoding order in advance, prior to generating each token.
> To address this limitation and enhance the applicability of dKV-Cache-Greedy, we propose several potential solutions:
>
> 1. **Decoding Position Estimation**
> Alternatively, we can design a lightweight prediction module to estimate the most likely position to decode in the next step, based on information from the previous iteration. By learning or heuristically approximating the next decoded position, we can use it instead of the random sampling method to achieve better performance and higher cacheability, with the $\mathcal{O}(N^2)$ complexity during inference.
>
> 2. **Relaxed Decoding Order via Window Tolerance**
> Rather than requiring the exact decoding position, we can instead predict an approximate decoding range, leveraging the inherent window mechanism in dKV-Cache-Greedy to provide tolerance for the imprecise prediction of the decoding position.  This relaxes the constraint from precise ordering to coarse localization.

---

> > ### Comment · Reviewer_TZvD · 2025-08-05
> > **Response to Authors**
> >
> > Thank you very much for the rebuttal. I think the effectiveness of dKV cache is still highly related to the decoding order. While existing dLLMs show better results with AR or near-AR decoding. Therefore, I think it still exists further exploration spaces regarding its effectiveness for truly random-ordered generation. Moreover, it still lags begind the AR models a lot in efficiency based on discussions with Reviewer vSQt
> >
> > Based on this new information, I decide to keep my score.

---

> > > ### Author Response · Authors · 2025-08-06
> > >
> > > We sincerely appreciate the reviewer’s thoughtful feedback on our rebuttal, which provides valuable guidance for further improving our work.

---

### Decision · Program_Chairs · 2025-09-17

**Decision:**

Accept (poster)

**Comment:**

This paper proposes a caching mechanism for diffusion language models (DLMs). As DLMs have emerged as an interesting LM alternative, inference efficiency becomes a critical issue and this paper is very timely in this regard. All but one reviewer find the proposed approach interesting and results solid. The authors did a great job during the rebuttal which helped further strengthen the work. After going through all the discussions, I believe that the concerns raised by reviewer 2PJU have been addressed by the authors, and it's unfortunate that the reviewer did not further engage in the discussions. I think this work would be very interesting to the DLM community and recommend accept.